# Explaining Hypergraph Neural Networks: From Local Explanations to Global Concepts

## Abstract

Hypergraph neural networks are a class of powerful models that leverage the message passing paradigm to learn over hypergraphs, a generalization of graphs well-suited to describing relational data with higher-order interactions. However, such models are not naturally interpretable, and their explainability has received very limited attention. We introduce SHypX, the first model-agnostic post-hoc explainer for hypergraph neural networks that provides both local and global explanations. At the instance-level, it performs input attribution by discretely sampling explanation subhypergraphs optimized to be faithful and concise. At the model-level, it produces global explanation subhypergraphs using unsupervised concept extraction. Extensive experiments across four real-world and four novel, synthetic hypergraph datasets demonstrate that our method finds high-quality explanations which can target a user-specified balance between faithfulness and concision, improving over baselines by 25 percent points in fidelity on average.

## 1 Introduction

Relational data in the form of graphs arises naturally in social networks (Fan et al., 2019), natural sciences (Zhang et al., 2021; Cranmer et al., 2019; Wang et al., 2021), traffic dynamics (Jiang & Luo, 2022), and knowledge databases (Schlichtkrull et al., 2018). The neural approach (Kipf & Welling, 2016) has enjoyed exciting successes, setting new state-of-the-art and expanding the reach of machine learning to new modalities (Ektefaie et al., 2023; Battaglia et al., 2018).

However, graphs can only describe pairwise relationships. This is insufficient to model real world systems that depend crucially on multi-way or group-wise interactions (Benson et al., 2016; Agarwal et al., 2005; Estrada & Rodríguez-Velázquez, 2006). A data structure that is well-suited to capturing higher-order correlations is the hypergraph. Whereas each edge in a graph joins two nodes, each hyperedge in a hypergraph joins an arbitrary number of nodes. Message passing principles extended to hypergraphs give rise to hypergraph neural networks (hyperGNNs) (Feng et al., 2019).

Unfortunately, graph neural networks (GNNs) and hyperGNNs share a key concern with all black-box neural models: their lack of explainability. In response, many post-hoc explainers (Ying et al., 2019; Luo et al., 2020; Yuan et al., 2021; Magister et al., 2021; Yuan et al., 2020) and interpretable-by-design architectures (Zhang et al., 2022b; Magister et al., 2023) have been developed for GNNs. However, the literature for hyperGNN explainability remains exceedingly sparse, with the hypergraph modality posing new challenges as the space of possible explanations is substantially larger than the graph counterpart.

In this work, we introduce SHypX(**S**ubhypergraph-based **Hyp**erGNN e**X**plainer), the first post-hoc hyperGNN explainer that produces explanations both at the instance level and global level. Our explanations take the form of subhypergraphs. Its core idea is to approximate subhypergraph sampling with a collection of independent Gumbel-Softmax samplers, and use gradient feedback from a loss function to obtain good explanation as per user specifications. This instance-level optimization is combined with concept extraction to produce global explanations, where concepts represent significant, recurring subhypergraphs. The design choices of our explainer are guided by several considerations: ensuring explanations are faithful to the hyperGNN under study, keeping explanations concise and legible, and avoiding the introduction of another black-box model in the explanation method.

To the best of our knowledge, this is the first global explainer designed for hypergraphs. For instance-level explanations, the only existing hypergraph explainer (Maleki et al., 2023) relies on learning an attention map to attribute the importance of each node-hyperedge link and induce the explanation subhypergraph. However, it remains contentious whether attention provides a valid explanation (Jain & Wallace, 2019; Wiegreffe & Pinter, 2019; Bibal et al., 2022). In contrast, SHypX is simple, effective, and doesn't rely on additional black-box networks to explain the hyperGNN.

In addition to introducing an effective hypergraph explainer, we also propose a set of synthetic datasets, designed to better assess the quality of hypergraph explanations along with suitable metrics for evaluation. As our experiments show, the current real-world datasets used in the previous work (Maleki et al., 2023) barely take into account the hypergraph structure, making it difficult to properly evaluate explainers. We believe that our datasets, which entirely depend on the higher-order structures, have the potential to speed up the advancements in the field of hypergraph explainability.

**Our main contributions** are summarized as follows:

1. We develop a **model-agnostic post-hoc explainer** for hyperGNNs that finds salient sub-hypergraphs **for both instance-level and global-level explanations**.

2. The **instance-level explainer alleviates the need for black-box attention mechanisms** used in the previous work. We integrate our instance-level explainer with unsupervised concept extraction to **produce a global-level explanation – a novelty in the field of hypergraph explainability**.

3. We introduce the first **hypergraph explainiability benchmark** containing four synthetic datasets which are highly structure-dependent and thus offer a challenging testbed for explainability. Moreover, we **generalize the fidelity metric** for explanation faithfulness, making it more sensitive to deviations induced by the explanation subhypergraph.

4. We conduct **extensive evaluations** on both synthetic and real-world datasets, showing that our explainer obtains coherent explanations for each class, outperforming existing methods.

## 2 RELATED WORK

**Hypergraph neural networks.** HyperGNNs operate over hypergraphs, taking inspiration from the message-passing paradigm of GNNs. HGNN (Feng et al., 2019), HyperGCN (Yadati et al., 2019), and HNHN (Dong et al., 2020) generalize GCN (Kipf & Welling, 2016) to hypergraphs. HCHA (Bai et al., 2021), HERALD (Zhang et al., 2022a), and HEAT (Georgiev et al., 2022) introduce attention mechanisms for hypergraphs to dynamically learn the incidence matrix, analogous to GAT (Veličković et al., 2017). UniGNN (Huang & Yang, 2021) proposes leveraging GNN architectures for updating node representations. AllSet (Chien et al., 2021) and EDHNN (Wang et al., 2023) use universal approximators to learn multiset functions for node and hyperedge updates. Our work proposes a model-agnostic explainer, producing hypergraph explanations regardless of the architectural choice.

**GNN explainers.** The majority of GNN explainers are local, finding an explanation subgraph pertaining to a specific input instance. Pope et al. (2019) and Sanchez-Lengeling et al. (2020) apply gradient-based attribution techniques from vision and language to graphs to produce local explanations. GNNExplainer (Ying et al., 2019) learns fractional edge weights and thresholds them to produce explanation subgraphs; this framework is extended by PGExplainer (Luo et al., 2020), which learns a second neural network to predict edge weights. SubgraphX (Yuan et al., 2021) finds the subgraphs instead by Monte Carlo Tree Search. GraphLIME (Huang et al., 2022) and PGMExplainer (Vu & Thai, 2020) learn explainable surrogates of the original GNN. In contrast, global explainers like XGNN (Yuan et al., 2020) and GCExplainer (Magister et al., 2021) produce explanations representative of a class: XGNN generates explanation graphs with policy gradients and GCExplainer with unsupervised concept extraction.

To the best of our knowledge, the only existing hyperGNN explainer is **HyperEX** (Maleki et al., 2023). It optimizes an attention-based network with InfoNCE to assign importance weights to node-hyperedge links to produce local explanations. However, there is ongoing debate about whether attention mechanisms offer valid explanations (Jain & Wallace, 2019; Wiegreffe & Pinter, 2019;

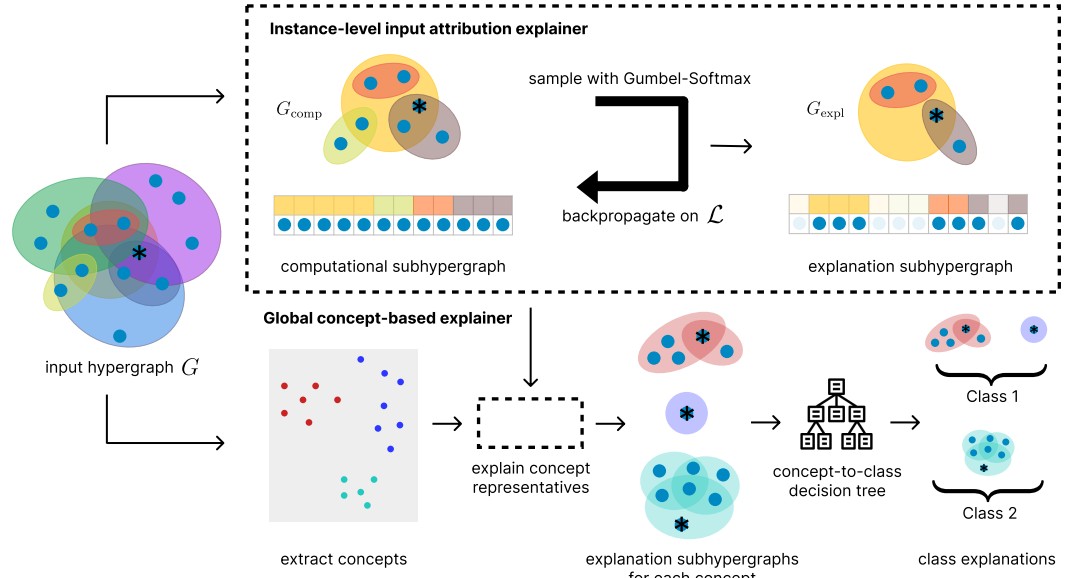

Figure 1: **Visualization of our hypergraph explainer providing local and global explanations.** (**Top**) Instance-level explanations are obtained by optimizing the subhypergraph structure using a loss function that incentivizes faithfulness (the explanation is able to reproduce the original prediction well) and concision (the explanation is as minimal as possible). (**Bottom**) Model-level explanations are obtained by combining the instance-level explainer with unsupervised concept extraction. After clustering the latent space into concepts, the closest node to each concept's center is picked as a representative and explained using the instance-level approach to produce concept and class-level explanations.

Bibal et al., 2022). In contrast, our model eliminates the need for surrogate networks, while also providing global-level explanations, a novelty in the realm of hypergraph explainability.

## 3 PRELIMINARIES

**Notation.** A hypergraph $G = (V, E)$ comprises a set of nodes $V$ and a set of hyperedges $E$. Each hyperedge $e = \{v_1, ..., v_{|e|}\} \in E$ is a set of nodes, and is said to be of degree $|e|$. In this sense, graphs are a special case of hypergraphs wherein all hyperedges have degree two. The structural content of a hypergraph is given by the incidence matrix $\boldsymbol{H} \in \mathbb{Z}_2^{|V| \times |E|}$, where $H_{ve} = \mathbb{1}(v \in e)$. $\boldsymbol{H}$ has an equivalent sparse representation as a hyperedge index of shape $(2, L)$, where $L = \sum_{e \in E} |e|$ is the number of node-hyperedge links and each column $[v, e]$ denotes that $v \in e$. The hypergraph has node features $\boldsymbol{X} = [\boldsymbol{x}_1, ..., \boldsymbol{x}_{|V|}] \in \mathrm{R}^{|V| \times d}$, where $d$ is the feature dimension and each $\boldsymbol{x}_i$ is associated to the node $v_i$.

Given a hypergraph $G = (V, E)$, we define a subhypergraph $G_{\text{sub}} = (V_{\text{sub}}, E_{\text{sub}})$ to be a subset $V_{\text{sub}} \subseteq V$ of the nodes, and a new set of edges $E_{\text{sub}}$ such that each $e_{\text{sub}} \in E_{\text{sub}}$ is a subset of precisely one hyperedge in the original hypergraph. Furthermore, we allow neither empty edges ($e_{\text{sub}} \neq \emptyset \; \forall \; e_{\text{sub}} \in E_{\text{sub}}$) nor isolated nodes ($\forall \; v \in V_{\text{sub}}, \; \exists \; e_{\text{sub}}$ such that $v \in e_{\text{sub}}$). Altogether, this can be thought of as taking a subset of columns of the hyperedge index.

**Problem statement.** Consider the task of node classification over a hypergraph. (Our explainer is more general, but we defer this discussion to Appendix A.) Let $f$ be a hyperGNN that outputs for each node $v$ a probability distribution $f(G, \boldsymbol{X}, v)$ over the classes. Our proposed model obtains **both local and global explanations** that are **architecture-agnostic** and **fully post-hoc**.

The goal of a *local* hypergraph explainer is, for each instance, to find which parts of the input hypergraph are most important to determining $f$'s output. Accordingly, the explanation artefact is a subhypergraph. A good explanation subhypergraph $G_{\text{expl}} = (V_{\text{expl}}, E_{\text{expl}})$ should be able to repro-

duce the original prediction well ("faithful") and also as minimal as possible ("concise"). Loosely speaking, we want $f(G, \boldsymbol{X}, v) \approx f(G_{\mathrm{expl}}, \boldsymbol{X}_{\mathrm{expl}}, v)$, where $\boldsymbol{X}_{\mathrm{expl}}$ is the restriction of $\boldsymbol{X}$ to $G_{\mathrm{expl}}$, for small $G_{\mathrm{expl}}$. While local explainers produce an explanation for each example, a *global* hypergraph explainer produces concise explanation subhypergraphs representative of each class.

# 4 METHOD

## 4.1 LOCAL EXPLAINER

Given a trained hyperGNN $f$, a hypergraph $G$, and a node instance $v$ in $G$, our goal is to produce an explanation subhypergraph that is both faithful and concise. To achieve this, we formulate these desiderata as a joint objective and optimize the explanation subhypergraph against this objective by discrete sampling. Figure 1(top) gives an overview of the local explainer.

**Objective function.** We can quantify the faithfulness of the explanation by the Kullback-Leibler divergence between the original class probabilities predicted by $f$ over $G$, and when $f$ is restricted to the explanation subhypergraph. We can quantify concision by the $L_1$ norm of the incidence matrix, which is equivalent to the number of node-hyperedge links. We denote this size measure on a hypergraph $G$ by $|G|_1$. These competing objectives suggest $G_{\mathrm{expl}} = \arg\min_{G_{\mathrm{sub}}} \mathcal{L}$, where the loss function is

$$\mathcal{L}(f, G_{\mathrm{sub}}, G, \boldsymbol{X}, v) = \lambda_{\mathrm{pred}} D_{\mathrm{KL}}\big(f(G_{\mathrm{sub}}, \boldsymbol{X}, v) \,||\, f(G, \boldsymbol{X}, v)\big) + \lambda_{\mathrm{size}}\big|G_{\mathrm{sub}}\big|_1, G_{\mathrm{sub}} \subseteq G, \quad (1)$$

and $\lambda_{\mathrm{pred}}$ and $\lambda_{\mathrm{size}}$ are hyperparameters governing the trade-off between faithfulness and concision.

For a message passing neural networks with $d$ layers, each node's receptive field is restricted to its $d$-hop neighborhood. This neighborhood defines a computation subhypergraph $G_{\mathrm{comp}} = (V_{\mathrm{comp}}, E_{\mathrm{comp}})$ which contains all information that determines the hyperGNN's output over that node. By simplifying the loss to

$$\mathcal{L}(f, G_{\mathrm{sub}}, G_{\mathrm{comp}}, \boldsymbol{X}, v) = \lambda_{\mathrm{pred}} D_{\mathrm{KL}}\big(f(G_{\mathrm{sub}}, \boldsymbol{X}, v) \,||\, f(G_{\mathrm{comp}}, \boldsymbol{X}, v)\big) + \lambda_{\mathrm{size}}\big|G_{\mathrm{sub}}\big|_1, G_{\mathrm{sub}} \subseteq G_{\mathrm{comp}}, \quad (2)$$

we reduce the search space of the explanation to a subhypergraph of $G_{\mathrm{comp}}$, which is typically much smaller than $G$.

**Optimization.** Exhaustively searching all $G_{\mathrm{sub}} \subseteq G_{\mathrm{comp}}$ is intractable due to the exponentially-large dimension of the search space. For a hypergraph with $n$ nodes and $m$ hyperedges of degree $d_1 \cdots d_m$, selecting a subhypergraph involves choosing from $2^{\sum_{i=1}^{m} d_i}$ potential subhypergraphs. In comparison, for a graph with $n$ nodes and $m$ edges, the number of possible subgraphs is much smaller ($2^m$), suggesting that finding the right explanation is particularly challenging in the hypergraph domain.

Instead, our approach is to optimize a joint probability distribution of the existence of each node-hyperedge link – in effect, a probability distribution over subhypergraphs – and obtain candidate subhypergraphs by discrete sampling. The sampler should be differentiable, admitting gradient updates to these probabilities. Note that our goal is to discretely optimize the structure of the $G_{\mathrm{sub}}$, and *not* the parameters of the hyperGNN, which remain fixed.

To ensure the sampler always produces a valid subhypergraph $G_{\mathrm{sub}}$, we impose the restriction that $\forall e_{\mathrm{sub}} \in E_{\mathrm{sub}}$ and $\forall v \in V_{\mathrm{sub}}$, $\Pr(v \in e_{\mathrm{sub}} = 0)$ if $v$ was not in the original, corresponding hyperedge of $G_{\mathrm{comp}}$. This ensures each $e_{\mathrm{sub}}$ is truly a subset of some hyperedge $e_{\mathrm{comp}} \in E_{\mathrm{comp}}$. Thus, our goal is to sample subhypergraphs from the joint distribution

$$\Pr(G_{\mathrm{sub}}) = \Pr(\{\mathbb{1}_{v \in e}\}_{\forall\, v \in V_{\mathrm{sub}},\, e \in E_{\mathrm{sub}}}), \quad v \notin e_{\mathrm{comp}} \implies \mathbb{1}_{v \in e} = 0, \quad (3)$$

where $\mathbb{1}$ is the indicator function.

We opt for a mean field approximation that decomposes the joint probability distribution into the product of marginals. Let $\pi_{v,e} := \Pr(\mathbb{1}_{v \in e} = 1)$. The approximation allows us to sample each node-hyperedge link independently:

$$\Pr(G_{\mathrm{sub}}) \approx \prod_{\forall v \in V_{\mathrm{sub}},\, e \in E_{\mathrm{sub}}} \pi_{v,e}, \quad v \notin e_{\mathrm{comp}} \implies \pi_{v,e} = 0. \quad (4)$$

Now we are faced with the problem of differentiable obtaining a discrete sample $y_{v,e}$ from the probabilities $\pi_{v,e}$ over each $v, e$ pair. We accomplish this using the Gumbel-Softmax (Jang et al., 2016; Maddison et al., 2016) over the binary categorical distribution described by $\pi_{v,e}$. The set of all incident node-hyperedge pairs $(v, e)$ such that $y_{v,e} = 1$ forms the explanation candidate $G_{\text{sub}}$.

We pass the resultant subhypergraph through the hyperGNN to evaluate $f(G_{\text{sub}}, \boldsymbol{X}, v)$. By ensuring this entire subhypergraph sampling is differentiable, we are able to optimize the underlying probabilities $\{\pi_{v,e}\}$, using backpropagation on the loss $\mathcal{L}(f, G_{\text{sub}}, G_{\text{comp}}, \boldsymbol{X}, v)$ defined in Equation 2.

**Post-processing.**  Following the approach described above, we extract the subhypergraph corresponding to the lowest loss observed during optimization. If this subhypergraph has disconnected components, we retain only the connected component containing the node $v$ being explained, and return it as the explanation $G_{\text{expl}}$. Disconnected components do not impact the hyperGNN output, so are typically pruned away by the size penalty in $\mathcal{L}$. However, this is not guaranteed due to the challenging loss landscape of this discrete problem. We discard disconnected components to produce a smaller and more legible explanation artefact, and grant the same advantage to the baselines in our evaluations.

## 4.2 GLOBAL EXPLAINER

The local explainer returns an explanation subhypergraph for a single node instance. How can we leverage this to obtain a global explanations at the class-level? While global explanation for hyperGNNs is an unexplored area of research, several methods were proposed for GNNs. However, creating class prototypes by graph alignment (Ying et al., 2019) is NP-hard, and graph generation with reinforcement learning (Yuan et al., 2020) requires expensive policy gradients optimization. We desire a global explainer whose computation costs do not scale with the increased combinatorial possibilities of the hypergraph space.

**Concept extraction and visualization.**  We propose to obtain global explanations using unsupervised concept extraction, inspired by Magister et al. (2021). Concepts are higher-level units of information, more accessible for humans than low-level neural network constructs (Ghorbani et al., 2019). Similar to the GNNs domain (Magister et al., 2021), we find that concepts may be identified with clusters in the hyperGNN's activation space. We then visualize each concept by finding the local explanation subhypergraph of its representative node.

Stated more precisely, a hyperGNN $f$ learns latent node representations $\boldsymbol{z}_v, \forall v \in V$. We train a $k$-means model with $k$ centroids on $\{\boldsymbol{z}_v\}_{v \in V}$, and use it to map each node $v$ onto one of $k$ concepts, $\texttt{KMeans}(\boldsymbol{z}_v) = c_v$. To obtain a concept-level explanation for concept $c$, we take the node closest to the cluster center,

$$v_c^* = \arg\min_{v:\, c_v=c} \left|\left| \boldsymbol{z}_v - (1/|c|) \sum_{u:\, c_u=c} \boldsymbol{z}_u, \right|\right| \tag{5}$$

where $|c|$ is the number of nodes belonging to that concept. We then produce as the explanation for concept $c$ the instance-level explanation subhypergraph for $v_c^*$, which we denote $G_{\text{expl}}(v)$. This explanation is computed using our instance-level explainer described in Section 4.1.

Figure 1(bottom) illustrates the overall pipeline. Whereas GCExplainer visualizes each concept by the $n$-hop graph neighborhood of $v_c^*$, where $n$ is a hyperparameter, the integration with our local explainer produces more legible explanation artefacts appropriate to the user's desired faithfulness-concision tradeoff (see Appendix D for a visual comparison between the two approaches).

**Explanation for each class.**  Users may desire explanations pertaining to each class. These explanations answer the question: what does a representative example of each class look like, according to the hyperGNN? To obtain such class-level explanations from our set of concept-level explanations, we use the majority vote function, $\texttt{MajorityVote} : \{c\} \to \{y\}$. That is, we take the most frequently occurring class of node instances belonging to a concept, and associate the concept with that class. The set of concepts associated with each class is taken as the explanation for that class:

$$\texttt{ClassExplanation}(y) = \left\{ G_{\text{expl}}(v_c^*) \right\}_{c:\, \texttt{MajorityVote}(c)=y}. \tag{6}$$

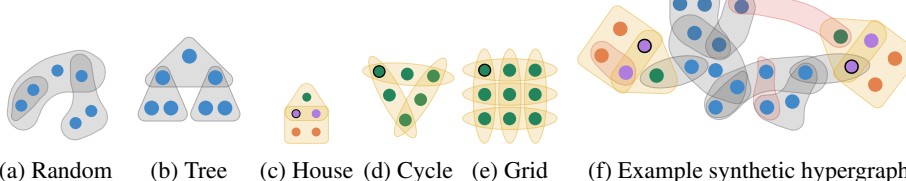

(a) Random    (b) Tree    (c) House (d) Cycle (e) Grid    (f) Example synthetic hypergraph

Figure 2: (a)-(b) Illustrative fragments of the "base" component of our synthetic hypergraphs. They come in two flavours: random, and tree (which is deterministic). (c)-(e) Synthetic hypergraph motifs of the house, cycle, and grid varieties. The node colors indicate class labels, which are each distinct from the class assigned to base nodes. The anchor node, whereby each motif is attached to the base, is denoted with a black outline. (e) A small example hypergraph of the H-RANDHOUSE family (pink edges denotes perturbations, gray denotes base hypergraph and yellow denotes attached motifs).

## 5 EXPERIMENTS

We show that our hypergraph explainer produces high quality explanations through extensive evaluations. We test on real hypergraphs CORA, COAUTHORCORA, COAUTHORDBLP, and ZOO from the benchmark of Chien et al. (2021).[1] In Section 5.1, we discuss why existing hypergraph datasets may not provide a sufficiently challenging setting for finding subhypergraph explanations, and design challenging synthetic hypergraph datasets to complement our evaluations. In Section 5.2, we highlight some shortcomings of the fidelity metric used to quantitatively evaluate explanations, and propose alternatives to address them.

### 5.1 SYNTHETIC HYPERGRAPHS

**Motivation.**    Synthetic graph datasets for GNN explainability have driven substantial progress in the field. However, no such dataset exists for hypergraphs. We argue that synthetic (hyper)graph datasets are valuable because they guarantee the primacy of structure for solving the task. For many real world hypergraphs like those in the benchmarks of Chien et al. (2021), competitive performance is already achieved by MLPs, which do not account for the hypergraph's structure. Accordingly, we find that node-level explanations obtained for such datasets typically comprise a "trivial" subhypergraph containing just the node itself. While valid explanations, they suggest the dataset fails to provide a challenging and discriminating testbed for evaluating hyperGNN explainability. Our synthetic hypergraphs ensure that labels depend critically on the hypergraph structure by construction, complementing evaluation on real world datasets.

**Dataset construction.**    Our synthetic hypergraphs are inspired by the synthetic graphs of Ying et al. (2019), which have served as a core benchmark in graph explainability. Each hypergraph comprises a "base" component that is either random or a deterministic "hyper-binary-tree" (Figure 2a-b), to which various "motifs" (Figure 2c-e) are attached using a single hyperedge. Additionally, we randomly add degree-2 hyperedges as perturbations. Figure 2e shows an example of a hypergraph constructed in this manner. The task is to classify nodes based on their positions in the base or motif. See Appendix B for details.

Different combinations of these base and motif components give rise to four synthetic hypergraphs: H-RANDHOUSE, H-COMMHOUSE, H-TREECYCLE, and H-TREEGRID. Table 3 shows their statistics and Table 4 benchmarks several hyperGNN architectures on these hypergraphs. Compared to benchmarks on real hypergraphs (Chien et al., 2021), our proposed datasets exhibits a clear gap between hyperGNNs and models that disregard structural information, such as MLPs. This indicates that the datasets represent challenging, structure-dependent tasks well-suited for evaluating hypergraph explainability.

---

[1]We selected the latter three hypergraphs because here the hyperGNNs outperform MLP by an appreciable margin; these are expected to be the relatively discriminating test cases for explainability, as discussed in Section 5.1 For comparison, we also selected CORA, where this is not the case.

## 5.2 METRICS

Evaluations of graph explainers often rely on comparison against the implanted motifs in synthetic datasets (Ying et al., 2019; Luo et al., 2020; Magister et al., 2021). Not only is this approach impossible for real world (hyper)graphs, due to the absence of reference motifs, we argue that it is unprincipled and potentially misleading. The implanted motifs reflect human reasoning, but are not necessarily faithful to the neural network, which may instead rely on a variant or correlate of the motif. Rather, a good explanation should provide users information about the hyperGNN's predictions, reflecting its internal mechanisms. This requirement is satisfied by the fidelity metrics (Amara et al., 2022):

$$\text{Fid}_- = 1 - \frac{1}{N} \sum_{i=1}^{N} \mathbb{1}(\hat{y}_i^{G_{\text{expl}}} - \hat{y}_i), \quad \text{Fid}_+ = 1 - \frac{1}{N} \sum_{i=1}^{N} \mathbb{1}(\hat{y}_i^{G_{\text{comp}} \setminus G_{\text{expl}}} - \hat{y}_i), \quad (7)$$

where $N$ is the number of instance-level predictions and $\hat{y}_i$ is the class prediction of the (hyper)GNN on the $i$th instance. The superscripts indicate a restriction of the (hyper)GNN to predict over that sub(hyper)graph. $G_{\text{comp}} \setminus G_{\text{expl}}$ is the complement sub(hyper)graph to the explanation sub(hyper)graph with respect to the computational sub(hyper)graph.[2] A low $\text{Fid}_-$ suggests the explanation is *sufficient*, and a high $\text{Fid}_+$ suggests the explanation is *necessary*. However, fidelity is vulnerable to some shortcomings, which we identify below and address with alternatives.

**Measuring faithfulness with generalized fidelity.** A major drawback of fidelity is that it is easily saturated. Because correct classification suffices to maximise each term in the sum, this metric is insensitive to more moderate perturbations to the logits. For example, we often care if the output class was predicted with 90% probability, or by only a narrow margin. To this end, we introduce a generalization to fidelity parametrized by a similarity function $s(\boldsymbol{p}, \boldsymbol{q})$, where $\boldsymbol{p}, \boldsymbol{q}$ are probability distributions over the classes $c \in C$:

$$\text{Fid}_-^s = \frac{1}{N} \sum_{i=1}^{N} s(\mathbf{p}_i^{G_{\text{expl}}}, \boldsymbol{p}_i), \quad \text{Fid}_+^s = \frac{1}{N} \sum_{i=1}^{N} s(\boldsymbol{p}_i^{G_{\text{comp}} \setminus G_{\text{expl}}}, \boldsymbol{p}_i). \quad (8)$$

Below we suggest a few good choices of $s$. Similar to a metric introduced by Agarwal et al. (2023), we can instantiate $s$ as the Kullback-Leibler divergence:

$$s_{\text{KL}}(\boldsymbol{p}, \boldsymbol{q}) := D_{\text{KL}}(\boldsymbol{p} \,\|\, \boldsymbol{q}) = \sum_{c \,\in\, C} p(c) \log\left(\frac{p(c)}{q(c)}\right). \quad (9)$$

The total variation distance is another apt statistical distance for our purpose. In a discrete probability space, it is essentially the L1 distance:

$$s_{\text{TV}}(\boldsymbol{p}, \boldsymbol{q}) := \frac{1}{2} \sum_{c \,\in\, C} \big| p(c) - q(c) \big|. \quad (10)$$

The negative cross-entropy is also a sensitive choice of $s$. It is equivalent to the logarithmic score, a strictly proper scoring rule in decision theory:

$$s_{\text{xent}}(\boldsymbol{p}, \boldsymbol{q}) := \sum_{c \,\in\, C} p(c) \log q(c). \quad (11)$$

Finally, the original fidelity metrics (Amara et al., 2022) are subsumed under this framework by choosing

$$s_{\text{Acc}}(\boldsymbol{p}, \boldsymbol{q}) := 1 - \mathbb{1}(\text{argmax}_c \, p(c) - \text{argmax}_c \, q(c)). \quad (12)$$

For regression tasks, $s$ can be replaced by MSE.

**Measuring concision with size.** By definition, a low $\text{Fid}_-^s$ shows that the explanation is faithful, since it can reproduce the original output over the full input hypergraph. But does a high $\text{Fid}_+^s$ indeed show that the explanation is also concise and *necessary*? We observe that $\text{Fid}_+^s$ can be especially misleading in the hypergraph context, since a subhypergraph's complement may also contain important nodes in $G_{\text{expl}}$. (Further discussion in Appendix E.) Instead, we propose to quantify concision

---

[2]The hypergraph complement is comprised of all the node-hyperedge links that exist in $G_{\text{comp}}$ but do not appear in $G_{\text{expl}}$. This generalizes the graph complement, which comprises the edges (and nodes at either end of the edge) which exist in $G_{\text{comp}}$ but do not appear in $G_{\text{expl}}$.

Table 1: **Quantitative evaluation of hyperGNN explainers on the synthetic benchmarks.** We compare explanation faithfulness, measured by generalized fidelity metrics, and concision, measured by subhypergraph size and density. Our method consistently outputs more faithful explanations than all baselines, which are given comparable or more generous size budgets ($n = 20$ for H-TREEGRID, $n = 10$ for all other datasets).

| | | $\mathrm{Fid}_-^{\mathrm{Acc}}$ ($\downarrow$) | $\mathrm{Fid}_-^{\mathrm{KL}}$ ($\downarrow$) | $\mathrm{Fid}_-^{\mathrm{TV}}$ ($\downarrow$) | $\mathrm{Fid}_-^{\mathrm{Xent}}$ ($\downarrow$) | Size ($\downarrow$) | Density ($\downarrow$) |
|---|---|---|---|---|---|---|---|
| H-RANDHOUSE | Random | 0.81 | 1.14 | 0.60 | 1.68 | 1.2 | 0.07 |
| | Gradient | 0.36 | 0.69 | 0.32 | 1.23 | 8.3 | 0.26 |
| | Attention | 0.61 | 0.82 | 0.45 | 1.36 | 3.6 | 0.17 |
| | HyperEX | 0.86 | 1.09 | 0.62 | 1.63 | 0.0 | 0.01 |
| | SHypX | **0.01** | **0.04** | **0.06** | **0.59** | 9.2 | 0.19 |
| H-COMMHOUSE | Random | 0.78 | 3.54 | 0.76 | 3.70 | 1.0 | 0.06 |
| | Gradient | 0.29 | 1.17 | 0.30 | 1.33 | 9.1 | 0.24 |
| | Attention | 0.71 | 3.03 | 0.70 | 3.19 | 1.6 | 0.09 |
| | HyperEX | 0.79 | 3.63 | 0.77 | 3.79 | 0.1 | 0.02 |
| | SHypX | **2e–3** | **0.02** | **0.03** | **0.18** | 9.2 | 0.20 |
| H-TREECYCLE | Random | 0.52 | 1.88 | 0.53 | 1.93 | 1.4 | 0.08 |
| | Gradient | 0.29 | 1.21 | 0.28 | 1.27 | 8.3 | 0.35 |
| | Attention | 0.26 | 0.91 | 0.31 | 0.97 | 3.0 | 0.16 |
| | HyperEX | 0.35 | 0.64 | 0.40 | 0.70 | 0.0 | 0.00 |
| | SHypX | **3e–3** | **0.01** | **0.01** | **0.07** | 5.6 | 0.22 |
| H-TREEGRID | Random | 0.68 | 2.11 | 0.63 | 2.30 | 8.6 | 0.35 |
| | Gradient | 0.40 | 1.04 | 0.36 | 1.24 | 17.9 | 0.56 |
| | Attention | 0.42 | 1.15 | 0.38 | 1.35 | 11.3 | 0.43 |
| | HyperEX | 0.66 | 1.63 | 0.57 | 1.82 | 13.4 | 0.46 |
| | SHypX | **0.01** | **0.02** | **0.04** | **0.22** | 15.1 | 0.45 |

by the size $|G_{\mathrm{expl}}|_1$ and density $|G_{\mathrm{expl}}|_1 / |G_{\mathrm{comp}}|_1$ of the explanation subhypergraph. We desire explanations of low size and low density. Density attains the maximum value of 1 iff $G_{\mathrm{expl}} = G_{\mathrm{compl}}$, in which case the explanation is perfectly (if trivially) faithful.

### 5.3 RESULTS

We compare our method against HyperEX (Maleki et al., 2023), which is currently the only hypergraph explainer in the literature, as well as Random, Gradient, and Attention baselines. (See Appendix C for further details on baselines and experimental setup.) For each dataset, all explanation methods are applied to the same model (a trained AllSetTransformer). Separately, we perform an ablation for our explainer's sampling technique in Section F.

**Synthetic hypergraphs.** Our method, SHypX, significantly outperforms all baselines across all four synthetic datasets (Table 1). While Gradient and Attention show substantial improvements from Random (e.g. on H-RANDHOUSE, $\mathrm{Fid}_-^{\mathrm{Acc}}$ is 0.36 and 0.61 respectively, compared to Random's 0.81), they don't consistently produce faithful explanations. On synthetic hypergraphs, HyperEX performs on par with Random. We hypothesize that this is because it mean-aggregates nodes to produce hyperedge representations, which constitutes a homophily assumption that is violated in the synthetic case. In comparison, the explanations produced by our method reliably achieves near zero fidelity metrics.

**Real hypergraphs.** On the real world hypergraphs, SHypX also outperforms all baselines. For example, in COAUTHOR-CORA, we achieve $\mathrm{Fid}_-^{\mathrm{KL}}$ of $3e-4$, compared to $0.03, 0.05, 0.08, 0.25$ for HyperEX, Gradient, Attention, and Random respectively. While producing more faithful explanations, our model does not sacrifice concision: it achieves this superior fidelity with the best concision on this dataset, at average size 2.1 and density 0.28. This relative ranking between methods is consistent across all four real hypergraphs. We also observe that the simple baselines Random, Gradient, and Attention already attain competitive performance on several real hypergraphs. CORA is the most extreme example of this, where even Random produces faithful explanations at $\mathrm{Fid}_-^{\mathrm{KL}} = 0.01$. Indeed, SHypX's mean explanation size of 1.4 suggests that oftentimes, just the node's features, without neighborhood structure, suffice to achieve perfect predictions over CORA. This "structural

Table 2: **Quantitative evaluation on four real world datasets.** Our method consistently produces explanations that are both more faithful (as measured by $\text{Fid}^*_*$ metrics) and more concise (as measured by Size and Density) than all baselines.

| | | $\text{Fid}^{\text{Acc}}_-$ ($\downarrow$) | $\text{Fid}^{\text{KL}}_-$ ($\downarrow$) | $\text{Fid}^{\text{TV}}_-$ ($\downarrow$) | $\text{Fid}^{\text{Xent}}_-$ ($\downarrow$) | Size ($\downarrow$) | Density ($\downarrow$) |
|---|---|---|---|---|---|---|---|
| CORA | Random | 0.01 | 0.01 | 0.01 | 0.05 | 3.7 | 0.90 |
| | Gradient | 0.01 | 0.03 | 0.01 | 0.06 | 3.9 | 0.91 |
| | Attention | 4e–3 | 0.02 | 0.01 | 0.05 | 3.7 | 0.91 |
| | HyperEX | 0.01 | 0.03 | 0.01 | 0.07 | 4.1 | 0.92 |
| | SHypX | **0.00** | **5e–4** | **1e–3** | **0.03** | 1.4 | 0.61 |
| COAUTHORCORA | Random | 0.10 | 0.25 | 0.09 | 0.31 | 5.4 | 0.67 |
| | Gradient | 0.01 | 0.05 | 0.02 | 0.11 | 7.2 | 0.74 |
| | Attention | 0.02 | 0.08 | 0.02 | 0.14 | 6.4 | 0.71 |
| | HyperEX | 0.01 | 0.03 | 0.02 | 0.10 | 7.4 | 0.75 |
| | SHypX | **0.00** | **1e–3** | **3e–3** | **0.07** | 2.1 | 0.28 |
| COAUTHORDBLP | Random | 0.11 | 0.48 | 0.14 | 0.48 | 5.5 | 0.52 |
| | Gradient | 0.01 | 0.03 | 0.01 | 0.03 | 8.4 | 0.60 |
| | Attention | 0.01 | 0.07 | 0.01 | 0.07 | 6.7 | 0.55 |
| | HyperEX | 0.01 | 0.05 | 0.01 | 0.05 | 8.8 | 0.61 |
| | SHypX | **0.00** | **3e–4** | **3e–4** | **2e–3** | 2.3 | 0.15 |
| ZOO | Random | 0.79 | 1.74 | 0.69 | 1.92 | 0.3 | 0.00 |
| | Gradient | **0.03** | 0.06 | 0.05 | 0.24 | 9.7 | 0.01 |
| | Attention | 0.08 | 0.26 | 0.08 | 0.44 | 3.1 | 0.00 |
| | HyperEX | 0.04 | 0.09 | 0.06 | 0.28 | 10.0 | 0.01 |
| | SHypX | **0.03** | **0.01** | **0.01** | **0.19** | 6.7 | 0.01 |

degeneracy" is also observed to some extent for COAUTHORCORA and COAUTHORDBLP. These results support Section 5.2's discussion about complementing evaluations on real hypergraphs with our challenging synthetic ones, and leveraging generalized fidelity as a more discriminating metric.

**Comparing explanation methods across different concision budget.** In Table 1 and Table 2, for each dataset, we fixed the same hyperparameter $n$ across all baselines (to obtain the top-$n$ node-hyperedge links) such that at least one baseline produces explanations of comparable concision to SHypX; this ensures a fair comparison between the fidelity results. We observe that the baseline explainers *often do not even select components that are connected* to the node being explained. Note that, since post-processing discards these disconnected components (see Section 4.1 ), $|G_{\text{expl}}| \leq n$ the explanation size can vary across baselines despite their identical choice of $n$. To understand how the quality of explanation varies when allowing larger subhypergraphs as explanation, we designed an experiment in which we directly control for the size of the final explanation and compare $\text{Fid}^{\text{KL}}_-$ (see Figure 3). The outperformance of our method is robust across the curve, whereas the baseline methods "buy" limited gains in fidelity with increasing size budget.

**Trading off faithfulness with concision.** By adjusting the relative strengths of the $\lambda_{\text{pred}}$ and $\lambda_{\text{size}}$ coefficients, our model allows the users to effectively trade off between explanation faithfulness and concision. Figure 3a shows H-RANDHOUSE explanations obtained with $\lambda_{\text{pred}}/\lambda_{\text{size}} \in \{0.2, 0.1, 0.05, 0.02, 0.01, 0.005\}$. As this ratio shrinks, the extracted explanations interpolate smoothly from concise-but-less-faithful (0.36 $\text{Fid}^{\text{KL}}_{-}$, mean size 4) to verbose-and-highly-faithful (3e−3 $\text{Fid}^{\text{KL}}_{-}$, mean size 22). Similarly, for ZOO, explanations obtained with $\lambda_{\text{pred}}/\lambda_{\text{size}} \in \{1e-2, 5e-3, 2e-3, 1e-3, 5e-4\}$ form a smooth decaying curve from higher to near-zero fidelity. Interestingly, Figure 3 suggests that for H-RANDHOUSE, all baselines perform similarly once adjusted for final explanation size, and that for ZOO, no baseline method reliably improves in fidelity with increasing size budget. Finally, we note that specifying the trade-off via $\lambda_{\text{pred}}/\lambda_{\text{size}}$ confers our method an additional benefit: it can dynamically adapt the explanation size for each node, according to the relevance of a node's neighborhood in the local hyperGNN prediction. In contrast, the baselines explainers inflexibly apply top-$n$ thresholding across all node instances.

**Global explanations.** Figure 4 shows the concept-level explanation subhypergraphs provided by our explainer for H-RANDHOUSE (for more datasets see Appendix D). We find that H-

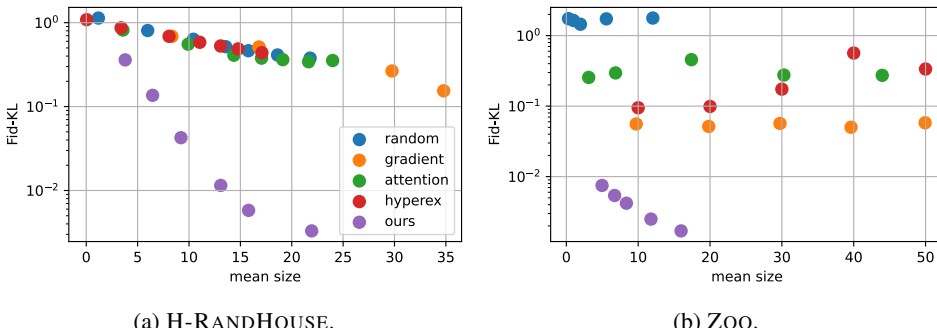

(a) H-RANDHOUSE.

(b) ZOO.

Figure 3: **Analysing the trade off between faithfulness and concision in various hypergraph explainers.** The figure shows $\text{Fid}^{\text{KL}}_{-}$ vs. mean explanation size for two select hypergraphs on two datasets. While all the baselines obtains very little improvement in fidelity as we increase the explanation size, our model consistently obtains more faithful explanations at every size budget.

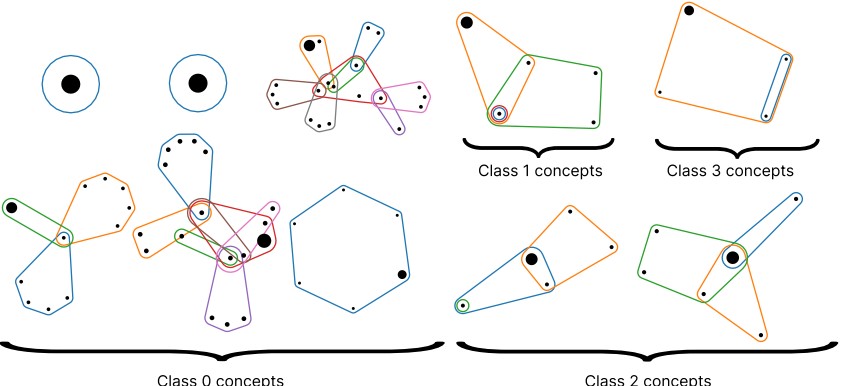

Figure 4: **Global concepts on H-RANDHOUSE dataset.** Class 0 is the base hypergraph, Class 1 is top-of-the-house, Class 2 is middle-of-the-house, and Class 3 is bottom-of-the-house. Concepts were extracted with 10 clusters, which sufficed to score well on the concept completeness metric (Appendix C.3.

RANDHOUSE's concept explanations are readily interpretable: the Class 1, 2, and 3 concepts clearly show each respective top-of-house, middle-of-house, and bottom-of-house node situated within a house-like motif. Particularly interesting is the subdivision of Class 2 into two distinct concepts: one for the "anchor node" that is attached to the base hypergraph (includes the attaching hyperedge), and one for the non-anchor node. This reveals that the hyperGNN implicitly represents and reasons about two types of Class 2 nodes. Furthermore, the Class 3 concept is visualized as a fragment of the house motif, suggesting that this hyperGNN does not rely on the top-of-house node to make Class 3 predictions. This mechanism is not a priori obvious, and such information could be leveraged to debug the hyperGNN. The remaining concepts corresponding to Class 0 reflect an eclectic variety, representative of the diverse neighborhoods of nodes in the random base graph.

## 6 CONCLUSION

Explainability for hyperGNNs is an under-explored topic, but essential for their responsible deployment in critical applications. We introduce SHypX, a model-agnostic post-hoc explainer, and demonstrate its efficacy with extensive evaluations. At the instance-level, our method finds explanation subhypergraphs that can target a desired tradeoff between explanation faithfulness and concision. At the model-level, we are the first to extend our instance-level method with concept extraction to efficiently derive concise global explanation subhypergraphs. Additionally, we design novel synthetic hypergraph datasets and propose more general fidelity metrics, which together allow for a challenging and sensitive evaluation of hyperGNN explainers.

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

# Appendix: Explaining Hypergraph Neural Networks: From Local Explanations to Global Concepts

This appendix contains details related to our proposed hypergraph neural network explainer as detailed below. The Supplementary Material also contains the full code associated with the proposed method.

- **Section A** contains a discussion about additional scenarios when our model can be applied, which where not fully explored in the main paper.
- **Section B** provides more details about the proposed synthetic benchmark.
- **Section C** contains implementation details about our model and the baselines used in our experiments.
- **Section D** highlights additional global-level visualizations and a qualitative comparison between the concepts extracted by GCExplainer and the one extracted by our model.
- **Section E** includes a detailed discussion about the limitations of $\text{Fid}_+$ metric.
- **Section F** includes an additional ablation study investigating the choice of sampling technique.

## A  EXTENSIONS AND DISCUSSION

**Beyond node classification.**  We focused on node classification tasks to simplify exposition. Nonetheless, our framework and methods is general to regression, as well as tasks that operate at the edge and graph level. In regression, we simply replace the KL divergence in $\mathcal{L}$ with MSE, since $f$ outputs regression targets instead of class probabilities. For hyperedge- and hypergraph-level tasks, $G_{\text{comp}}$ may be more complex than a $d$-hop neighborhood, depending on the architecture, or even equal to $G$. Additionally, we allow disconnected components if they contribute to the final prediction, e.g. for hypergraph-level tasks formulated with a global pooling layer. The overall pipeline is otherwise unchanged.

**Feature selection.**  The novelty of our hypergraph explainer lies in "structure selection", i.e. finding the explanation subhypergraph. We may wish to simultaneously find a subset of the features which are most important to each local instance. Ying et al. (2019) accomplishes feature selection by learning a $L_1$-regularized mask $M$ over feature vectors, which we may directly adopt to update our objective to

$$\mathcal{L}\big(f, G_{\text{sub}}, G_{\text{comp}}, \boldsymbol{X}, \boldsymbol{M}, v\big) = \lambda_{\text{pred}} D_{\text{KL}}\big(f(G_{\text{sub}}, \boldsymbol{X}, v) \,||\, f(G_{\text{comp}}, \boldsymbol{X} \odot \boldsymbol{M}, v)\big)$$
$$+ \lambda_{\text{size}}\big|G_{\text{sub}}\big|_1 + \lambda_{\text{feat}}\big|\boldsymbol{M}\big|_1, \quad G_{\text{sub}} \subseteq G_{\text{comp}}, \quad G_{\text{expl}}, \boldsymbol{M}^* = \underset{G_{\text{sub}}, \boldsymbol{M}}{\arg\min}\, \mathcal{L}. \quad (13)$$

Since this approach is equally suitable for graphs and hypergraphs (and indeed, any other modality with multidimensional features), we do not focus on it in the present work.

**Generality across architectures.**  Finally, we emphasize that SHypXis model-agnostic. It only relies on the high level message passing abstraction, which ensures that the notion of a computational subhypergraph is well-defined, and $f$ can accept any subhypergraph as an input. Our explainer can be applied to any hyperGNN, such as HGNN, HCHA, UniGNN, and AllSet models.

## B  SYNTHETIC DATASET

Our synthetic hypergraphs are designed with a "base-and-motif" construction , inspired by Ying et al. (2019). For the random base, we sample a random bipartite graph with $n, m$ nodes in each of the bipartite sets respectively, and $k$ edges between them uniformly at random. We take the largest

Table 3: Construction of novel synthetic hypergraphs. Upper section reports fundamental properties of each hypergraph dataset, such as its base and type of attached motif. Lower section reports, for each family, the default parameters used to instantiate the hypergraph used in our evaluations.

|  | H-RANDHOUSE | H-COMMHOUSE | H-TREECYCLE | H-TREEGRID |
|---|---|---|---|---|
| base | random | random | tree | tree |
| motif | house | house | cycle | grid |
| node feat. | none (ones) | bimodal normal | none (ones) | none (ones) |
| # classes | 4 | 8 | 2 | 2 |
| # base nodes | 312 | 648 | 255 | 255 |
| # motifs | 100 | 200 | 80 | 80 |
| # perturb. edges | 80 | 80 | 80 | 80 |
| # inter-community edges | - | 80 | - | - |

connected component of this bipartite graph and apply the inverse star expansion to obtain a random base hypergraph (Figure 2a). For the tree base, we enclose each triplet of a parent node and its two child nodes in a hyperedge. This produces a tree base hypergraph that is deterministic and 3-uniform (Figure 2b). The house, cycle, and grid motifs from Ying et al. (2019) are also lifted to hypergraph motifs (Figure 2c-e). In designing these, we were motivated by preserving the natural symmetries of each motif, without rendering the classification task trivial (for example, allowing motifs to be immediately distinguishable from a tree base by hyperedge degree). In the example visualized in Figure 2e, the hypergraph consists of a random base of 13 nodes (blue nodes and grey hyperedges), 2 house motifs, and 3 edge perturbations (pink hyperedges).

Different combinations of these base and motif components give rise to the synthetic hypergraphs H-RANDHOUSE, H-COMMHOUSE, H-TREECYCLE, and H-TREEGRID (Table 3). H-COMMHOUSE comprises two H-RANDHOUSE graphs, i.e. "communities", stitched together with random edges. Each node has features drawn from a normal distribution, whose mean and variance depend on the community membership. The other three synthetic graphs have trivial features, which we choose to be all ones. (We observed similar performance for all zeros or standard random normal features.) Perturbations, in the form of degree-2 hyperedges, are then added randomly to simulate structural noise, increasing the difficulty of the task. A train-validation split at 80% train nodes is applied to each hypergraph.

The benchmark task over our synthetic hypergraphs is node classification, where the node labels depend on the node's position in the base or motif. Each class is denoted by a distinct color in Figure 2. In particular, all base nodes are Class 0, and all nodes in the cycle and grid motif are Class 1. The house motif is further sub-divided into top-of-the-house (Class 1), middle-of-the-house (Class 2), and bottom-of-the-house (Class 3).

We benchmark several hyperGNN architectures on our synthetic tasks. As claimed, the synthetic hypergraphs are challenging. Table 4 shows that performance improves with stronger models, and the structure-agnostic MLP does no better than random.

Table 4: Benchmarking hypergraph neural networks on the synthetic hypergraphs. Each number denotes the mean final validation accuracy, in %, over 5 random seeds. All models are three layers deep, use sum aggregation, and no dropout. AllDeepSets and AllSetTransformer have dimension-16 message passing and classifier layers; MLP, HGNN, HCHA have dimension-80 hidden layers, which ensures all models have comparable parameter count. All models are trained with the Adam optimizer at 0.001 learning rate, for 2000 epochs (MLP, HGNN, HCHA) or 500 epochs (AllDeepSets and AllSetTransformer), which we observed sufficed to achieve convergence. Other hyperparameters are per Chien et al. (2021)'s defaults. Boldface indicates the best model.

|                  | H-RANDHOUSE | H-COMMHOUSE | H-TREECYCLE | H-TREEGRID |
|------------------|-------------|-------------|-------------|------------|
| MLP              | $38.65_{\pm 0.00}$ | $28.91_{\pm 2.39}$ | $65.31_{\pm 0.00}$ | $73.85_{\pm 0.00}$ |
| HGNN             | $79.75_{\pm 10.34}$ | $60.30_{\pm 1.59}$ | $85.44_{\pm 2.57}$ | $92.62_{\pm 2.80}$ |
| HCHA             | $56.32_{\pm 20.48}$ | $26.12_{\pm 10.47}$ | $65.31_{\pm 0.00}$ | $78.26_{\pm 9.58}$ |
| AllDeepSets      | $89.20_{\pm 7.18}$ | $93.33_{\pm 9.87}$ | $\mathbf{86.26}_{\pm 9.09}$ | $87.49_{\pm 4.39}$ |
| AllSetTransformer | $\mathbf{95.09}_{\pm 6.95}$ | $\mathbf{97.15}_{\pm 2.29}$ | $83.95_{\pm 12.38}$ | $\mathbf{90.05}_{\pm 4.79}$ |

## C    FURTHER EXPERIMENT DETAILS

### C.1    BASELINES

We compare our explainer against four baseline methods: Random, Gradient, Attention, and HyperEX. Each of these baselines is parametrized by $n$, such that the top-$n$ node-hyperedge links are selected according to each method's importance ranking.

- **Random**. The importance score of each node-hyperedge link is randomly assigned as a random variable drawn from $U(0,1)$. That is, we get the subhypergraph induced by $n$ uniformly random node-hyperedge links.

- **Gradient**. We compute the gradient of the logit on the predicted class of the node being explained, with respect to the hypergraph edge index. We then get the subhypergraph induced by the $n$ node-hyperedge links with the largest non-zero gradients by absolute value.

  Note that our gradient baseline is significantly more competitive than the ostensibly similar saliency and integrated gradients baselines constructed by Maleki et al. (2023). Our gradient baseline computes gradients on the hyperedge index, and thus selects a set of node-hyperedge links. Their gradient baselines compute gradients over the input nodes, and thus selects a set of nodes.

- **Attention**. This baseline is only feasible for hyperGNNs with an attention mechanism. Since we produce all explanations with respect to the AllSetTransformer architecture, this is satisfied. We compute the mean of the attention weights from each layer. For each AllSetTransformer layer, this includes attention weights in both the node-to-hyperedge and hyperedge-to-node directions. We then get the subhypergraph induced by the $n$ node-hyperedge connections with the largest non-zero attention weights by absolute value.

- **HyperEX**. The hypergraph explainer by Maleki et al. (2023) proposes to calculate importance weights between nodes and hyperedges with a shallow attention model surrogate parametrized as

$$\alpha_{ve} = \frac{\exp(\omega_{ve})}{\sum_{\tilde{e}:v\in\tilde{e}} \exp \omega_{v\tilde{e}}}, \quad \omega_{ve} = (\boldsymbol{W}_Q \boldsymbol{z}_v)^T \cdot (\boldsymbol{W}_K \boldsymbol{h}_e) \cdot s_v, \qquad (14)$$

  where $\boldsymbol{W}_Q, \boldsymbol{W}_K, s_v$ are learnable weights. $\boldsymbol{z}_v$ is the latent representation of node $v$, per the trained hyperGNN, and $\boldsymbol{h}_e$ is the latent representation of hyperedge $e$, which Maleki et al. (2023) compute by mean aggregation of its neighborhood: $\boldsymbol{h}_e = \frac{1}{|\mathcal{N}(e)|} \sum_{v\in\mathcal{N}(e)} \boldsymbol{z}_v$.

  HyperEX's code is not publicly released, but was shared with us in private communications. To facilitate fair comparison with our method and other baselines, we adapt their implementation into our own pre- and post-processing pipelines. Like the original authors, we choose the hidden dimension of the attention surrogate model to be 16 and train on 50% of the node instances with InfoNCE loss. We choose the learning rate 0.1 by hyperparameter search. HyperEX requires retraining a new model for each choice of $n$.

Table 5: Task performance (accuracy on train, validation, and test splits) for each dataset, and the concept completeness of extracted concepts. (Note we did not use a separate test split for the synthetic datasets in our experiments.) The decision tree classifier used to compute concept completeness uses the same train/validation split as the base task.

| | Train Acc. | Val Acc. | Test Acc. | $k$ | Concept completeness |
|---|---|---|---|---|---|
| H-RANDHOUSE | 0.98 | 0.96 | - | 10 | 0.96 |
| H-COMMHOUSE | 1.00 | 1.00 | - | 15 | 0.96 |
| H-TREECYCLE | 0.99 | 0.98 | - | 10 | 0.98 |
| H-TREEGRID | 0.96 | 0.96 | - | 10 | 0.94 |
| CORA | 1.00 | 0.79 | 0.77 | 10 | 0.72 |
| COAUTHORCORA | 1.00 | 0.84 | 0.82 | 10 | 0.87 |
| COAUTHORDBLP | 1.00 | 0.91 | 0.91 | 10 | 0.93 |
| ZOO | 1.00 | 0.96 | 0.96 | 10 | 1.00 |

For the baselines in each dataset, we choose $n$ such that the density of the gradient or attention explanations is comparable to, or greater than, the density of our explanations. This ensures our method does not have an unfair advantage. We find that $n = 10$ for all datasets except $n = 20$ for H-TREEGRID suffices to achieve this comparison. Note that these size budgets are greater than the mean size of explanations produced by our method on their respective datasets. Alternatively, Figure 3 compares all methods across the entire curve of varying explanation size budgets.

For consistency, all explanation methods operate over the same AllSetTransformer model for each dataset. This model's task performance is reported in Table 5. All explanation methods benefit from identical pre-processing, which reduces the search space to the computational subhypergraph. They are also subject to the same post-processing, which retain only the connected component containing the node being explained (as described in Section 4.1). This means the mean explanation size obtained is generally less than $n$.

## C.2 HYPERPARAMETERS FOR SHYPX

For the main results of Table 1 and Table 2, we choose our explanation concision budget by setting $\lambda_{\text{pred}} = 1$, and $\lambda_{\text{size}} = 0.05$ for the synthetic datasets and $\lambda_{\text{size}} = 0.005$ for the real world datasets. Alternative choices of $\lambda$ for two select datasets are reported in Figure 3. The explanation subhypergraph is sampled with Gumbel-Softmax at temperature 1.0, and optimized with Adam for 400 epochs at learning rate 0.01. The probability of sampling each node-hyperedge link ($\pi_{v,e}^{(1)}$) is initialized uniformly to $\approx 95\%$ across the computational subhypergraph.

## C.3 CONCEPT EXTRACTION AND CONCEPT COMPLETENESS

We extract concepts by k-means clustering, as described in Section 4.2. The quality of concept extraction is quantified by concept completeness, the accuracy of a decision tree classifier that optimally maps the set of concepts onto the set of class labels (Magister et al., 2021). Optimal is defined such that each node instance, featurized only by its concept label, is mapped to a class label with high accuracy. Since the concept label is a discrete class, the decision tree classifier is optimized by performing majority vote within each concept, as proposed in Section 4.2. We consider the concept extraction successful if its concept completeness is close to the task accuracy, since this overall procedure relies on the latent representations learned by the hyperGNN.

Table 5 shows that across all datasets, the latents are indeed such that we can successfully extract meaningful concepts that score well on concept completeness (i.e. within a few percentage points of the task accuracy). We find that $k = 10$ suffices to achieve this condition on all datasets, but that it is beneficial to increase to $k = 15$ for H-COMMHOUSE.

# D  FURTHER CONCEPT VISUALIZATIONS

## D.1  CONCEPTS FOR OTHER HYPERGRAPHS

We report concept visualizations for H-COMMHOUSE (Figure 5), H-TREECYCLE (Figure 6), and H-TREEGRID (Figure 7), analogous to Figure 4 for H-RANDHOUSE.

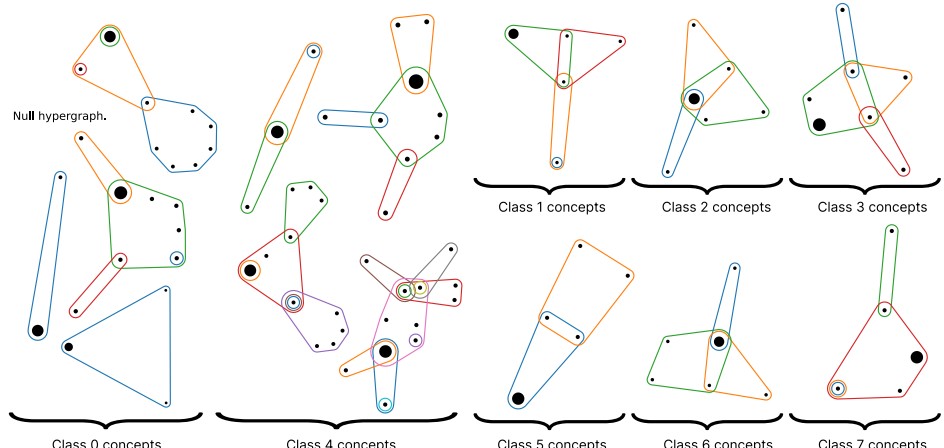

Figure 5: Concepts for H-COMMHOUSE.

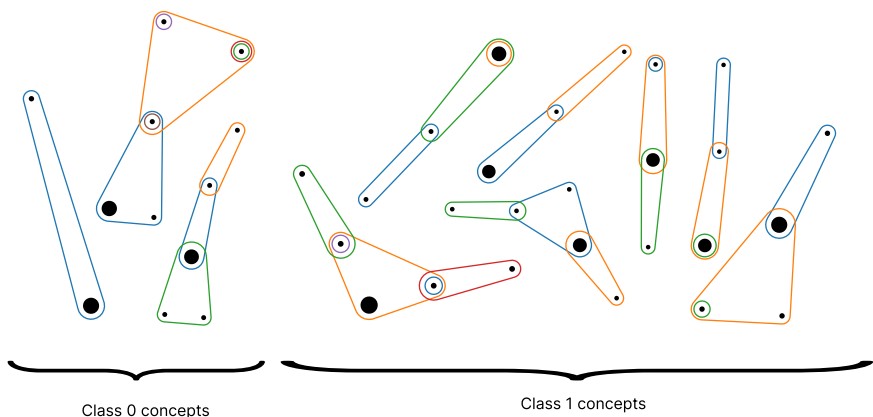

Class 0 concepts            Class 1 concepts

Figure 6: Concepts for H-TREECYCLE.

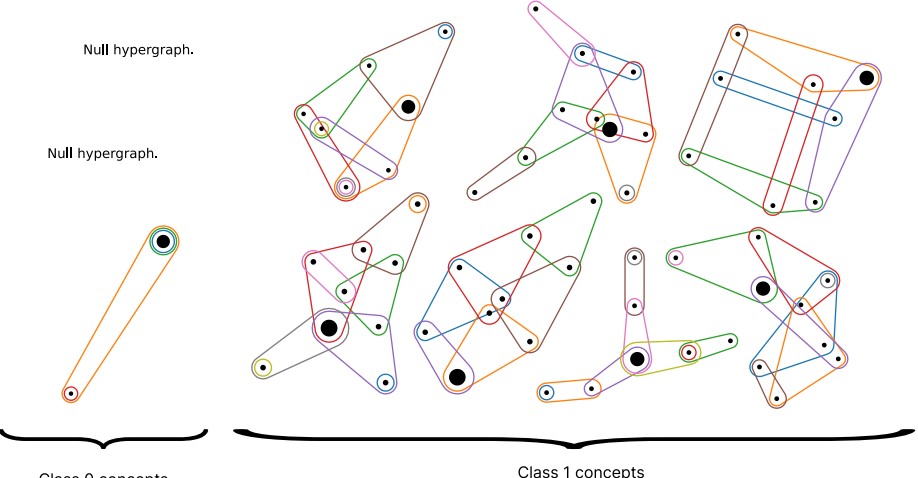

Class 0 concepts            Class 1 concepts

Figure 7: Concepts for H-TREEGRID.

## D.2 IMPROVEMENT OVER GCEXPLAINER

Visualizing concepts by the $n$-hop neighborhood of their representative nodes, as suggested by directly generalizing the GNN explainer of Magister et al. (2021), can produce crowded hypergraphs that obscure the crucial neighborhood important to that node instance. In Figure 8 and Figure 9 for H-RANDHOUSE and COAUTHORCORA respectively, we demonstrate with a few examples of concepts extracted from each hypergraph. For H-RANDHOUSE, we see that our method (bottom row) more clearly reveals the house motif when explaining nodes located in the motif. For COAUTHORCORA, the frequent appearance of the trivial subhypergraph (i.e., comprising only the node being explained) in our explanations reveals that class labels depend more strongly on features than local structure. This observation is not apparent from visualising $n$-hop neighborhoods (top row).

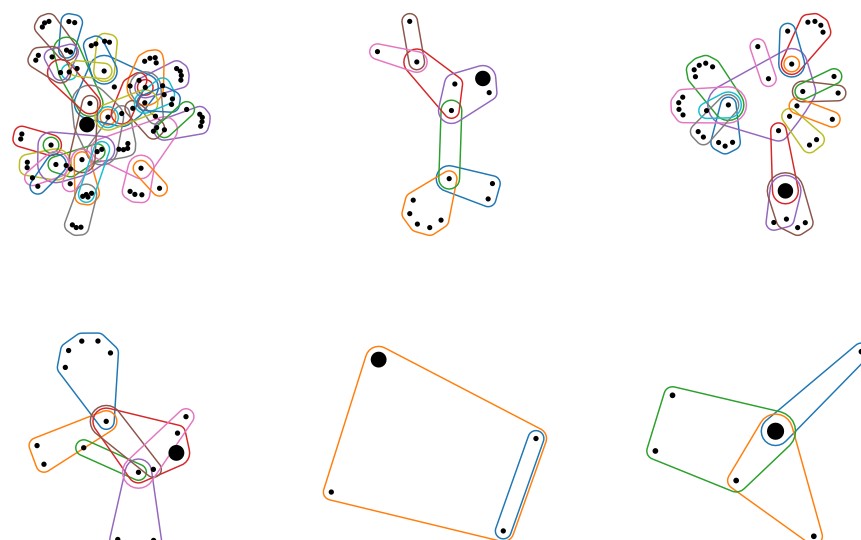

Figure 8: Concepts visualized by the $n$-hop expansion (setting $n = 3$, the depth of the hyper-GNN) for COAUTHORCORA (top row), and their respective visualizations when simplified using our method (bottom row). First column: Class 0 node from base. Second column: Class 3 node from house motif. Third column: Class 2 node from house motif.

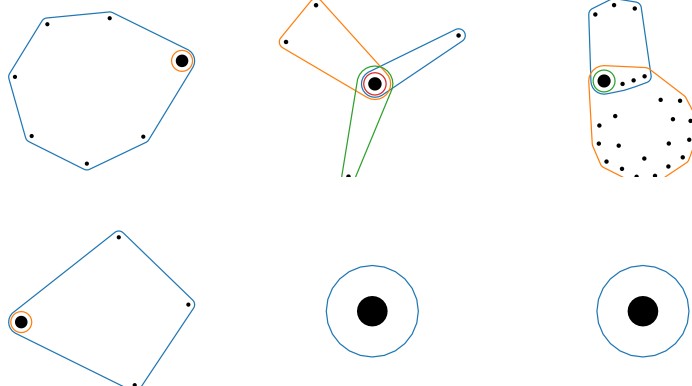

Figure 9: Concepts visualized by the $n$-hop expansion (setting $n = 1$, the depth of the hyperGNN) for H-RANDHOUSE (top row), and their respective visualizations when simplified using our method (bottom row).

# E   LIMITATIONS OF FID+

The $\text{Fid}_+$ metric has been used to measure whether an explanation is "sufficient", that is, whether it is free of superfluous information. A large $\text{Fid}_+$ indicates that the explanation's complement *does not contain* useful information for the hyperGNN's prediction. This has been thought to suggest that the explanation has successfully isolated the useful information. However, this reasoning is flawed – a successful explanation (achieving the user-desired balance of faithfulness and concision) could nonetheless induce a complement subhypergraph that can also reproduce the hyperGNN's prediction. This can be seen with a simple intuition: when the explanation subhypergraph is concise – that is, all of its parts are necessary, as desired – the complement is large. The complement is therefore likely to include a large number of hyperedges and neighbors directly incident to the node

Table 6: Fidelity and size metrics on the explanation complement, for two select datasets. We find that this can be a misleading metric.

| | | $\text{Fid}_+^{\text{Acc}}$ ($\uparrow$) | $\text{Fid}_+^{\text{KL}}$ ($\uparrow$) | $\text{Fid}_+^{\text{TV}}$ ($\uparrow$) | $\text{Fid}_+^{\text{Xent}}$ ($\uparrow$) | Size ($\uparrow$) | Density ($\uparrow$) |
|---|---|---|---|---|---|---|---|
| H-TREEGRID | Random | 0.59 | 1.86 | 0.55 | 2.06 | 29.3 | 0.65 |
| | Gradient | 0.73 | 2.23 | 0.67 | 2.43 | 30.6 | 0.78 |
| | Attention | 0.42 | 1.36 | 0.39 | 1.56 | 33.1 | 0.80 |
| | HyperEX | **0.77** | 2.23 | **0.70** | 2.42 | 24.4 | 0.54 |
| | SHypX | **0.77** | **2.29** | **0.70** | **2.48** | 22.8 | 0.55 |
| COAUTHORDBLP | Random | 0.32 | 0.95 | 0.33 | 0.95 | 22.2 | 0.48 |
| | Gradient | 0.56 | 1.75 | 0.61 | 1.75 | 19.3 | 0.40 |
| | Attention | 0.47 | 1.52 | 0.49 | 1.52 | 21.0 | 0.45 |
| | HyperEX | **0.72** | **2.25** | **0.80** | **2.25** | 18.9 | 0.39 |
| | SHypX | 0.07 | 0.33 | 0.09 | 0.34 | 25.4 | 0.85 |

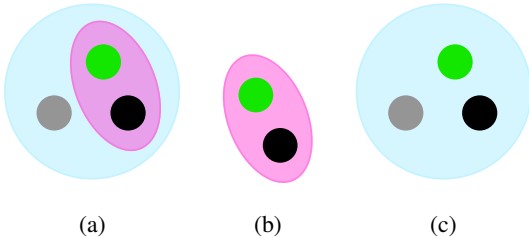

(a)  (b)  (c)

Figure 10: A minimal example demonstrating how $\text{Fid}_+$ can be prone to undesirable behavior. (a) In this hypergraph, we wish to locally explain a hyperGNN's output over the black node. The green node provides perfect information about the class of the black node, while the grey node is irrelevant. (b) This explanation subhypergraph is small and can faithfully reproduce the output over the black node. These desirable qualities are reflected in its size $\text{Fid}_-$ metrics (both low). (c) However, its complement also contains the important green node, resulting in a poor (low) $\text{Fid}_+$ score, despite the apparent optimality of the explanation.

being explained. This allows the complement to reproduce the hyperGNN's prediction with high fidelity, producing low $\text{Fid}_+$ scores. See Figure 10 for an illustrative example.

To concretely illustrate some of these failure modes, we expose the $\text{Fid}_+$ scores of H-TREEGRID and COAUTHOR-DBLP in Table 6. For H-TREEGRID, the similar $\text{Fid}_+$ for Gradient and our method suggest that they are comparably successful at isolating relevant information to the explanation sub-hypergraph. However, this does not align with our natural understanding of which explanations are more "sufficient" – whereas Table 1 shows that our method achieves an extremely low fidelity at mean explanation size of 15.1 and mean explanation density of 0.45, Gradient produces explanations with $\text{Fid}_-^{Acc} = 0.40$ while being almost 3 links larger and 10 percentage points denser. For COAUTHORDBLP, our method yields the most faithful and most concise explanations (average size 2.3 and average density 0.15) of all baselines (Table 2). However, the small size of these explanations induces a large complement, contributing to its unfavorable $\text{Fid}_+$ scores.

Based on these observations, we opt for hypergraph size $|G|_1$ (Section 5.2) as a cheaper and less artefact-prone measure of explanation minimality.

## F  SAMPLER ABLATION

In this section, we investigate the choice of sampling technique. Since this choice pertains to the optimization, we are primarily interested in which sampler achieves the lowest loss given a fixed objective function (Equation 2). Table 7 compares the loss attained by the Gumbel-Softmax sampler (our choice) against two alternatives, as well as reporting their respective fidelity and size metrics for reference.

Table 7: Ablating the choice of Gumbel-Softmax sampler to two alternatives: relax-and-thresh (Ying et al., 2019) and sparsemax (Martins & Astudillo, 2016). Here, the loss function has coefficients $\lambda_{\text{pred}} = 1$ and $\lambda_{\text{size}} = 0.005$. Lowest losses are in boldface.

| | | Loss ($\downarrow$) | Fid$_-^{\text{Acc}}$ ($\downarrow$) | Fid$_-^{\text{KL}}$ ($\downarrow$) | Fid$_-^{\text{TV}}$ ($\downarrow$) | Size ($\downarrow$) | Density ($\downarrow$) |
|---|---|---|---|---|---|---|---|
| H-RANDHOUSE | gumbel-softmax | **0.10** | 0.00 | 0.00 | 0.01 | 19.5 | 0.32 |
| | relax-and-thresh | 0.15 | 0.07 | 0.06 | 0.06 | 18.2 | 0.31 |
| | sparsemax | 0.58 | 0.28 | 0.52 | 0.25 | 12.5 | 0.21 |
| ZOO | gumbel-softmax | **0.04** | 0.03 | 0.01 | 0.01 | 6.7 | 0.01 |
| | relax-and-thresh | 0.14 | 0.07 | 0.09 | 0.06 | 10.4 | 0.01 |
| | sparsemax | 0.08 | 0.05 | 0.04 | 0.04 | 6.5 | 0.01 |

We compare against a "**relax-and-thresh**" method, which is the continuous relaxation for GNN explanations popularized by GNNExplainer (Ying et al., 2019). Relax-and-thresh learns real-valued probability weights over the incidence matrix, which are optimized by gradient descent. To encourage discrete sampling, it employs an entropy penalty to softly regularize these weights to 0s and 1s. Discreteness is only strictly enforced during post-processing: after optimization, the probability weights are binarized by thresholding (typically at 0.5) to produce the explanation subhypergraph. (Upon binarization, the entropy loss becomes trivially zero.)

We also try replacing Gumbel-Softmax with a **sparsemax** sampler (Martins & Astudillo, 2016). Whereas the familiar softmax function maps logits $z_i$ onto a probability distribution by $\text{softmax}_i(\boldsymbol{z}) = \exp(z_i) / \sum_j \exp(z_j)$, sparsemax proposes an alternative transformation:

$$\text{sparsemax}(\boldsymbol{z}) = \underset{\boldsymbol{p} \in \Delta^{K-1}}{\arg\min} \, ||\boldsymbol{p} - \boldsymbol{z}||^2, \tag{15}$$

where $\Delta^{K-1} = \{\boldsymbol{p} \in \mathbb{R}^K \mid \mathbf{1}^T \boldsymbol{p} = 1, \boldsymbol{p} \geq \mathbf{0}\}$ is the $(K-1)$-dimensional simplex. Whereas the softmax probability distribution has full support, the sparsemax probability distribution is likely to be sparse. This is because it is the Euclidean projection of $\boldsymbol{z}$ onto the probability simplex and is likely to hit the boundary. For fair comparison, we also optimize with an entropy loss term during sparsemax sampling, and binarize post-optimization.

We performed this ablation for one synthetic (H-RANDHOUSE) and one real (ZOO) dataset. Table 7 shows that Gumbel-Softmax achieves better losses than both relax-and-thresh and sparsemax: 0.10 (vs 0.15 and 0.58) on H-RANDHOUSE, and 0.04 (vs 0.14 and 0.08) on ZOO. Even without reference to the quantitative results, we know that relax-and-thresh and (to a lesser extent) sparsemax suffer from the so-called "introduced evidence problem" (Dabkowski & Gal, 2017; Yuan et al., 2022). Because the weighted subhypergraph seen during optimization differs from the final explanation subhypergraph obtained upon binarization, these samplers can lead to highly unfaithful explanations. Though relax-and-thresh attempts to mitigate this effect with entropy loss, we find that it is insufficient to avoid this problem, particularly for hypergraphs. The sparsity properties of sparsemax make it less prone to this failure mode (it achieves a much higher rate of zero entropy loss), but does not eliminate the problem completely. Note that HyperEX (Maleki et al., 2023) is also prone to the "introduced evidence problem", since it also thresholds attention weights to obtain the final explanation subhypergraph.

