# OpenReview forum: "Explaining Hypergraph Neural Networks: From Local Explanations to Global Concepts"
_ICLR.cc/2025/Conference — Submitted to ICLR 2025_

### Official Review · Reviewer_agpw · 2024-10-25

**Soundness:** 2
**Presentation:** 2
**Contribution:** 2
**Rating:** 3
**Confidence:** 4

**Summary:**

This paper proposes an explainer for hyperGNNs at both instance-level and global-level, which is called SHypX. At both instance-level and global-level, they extract salient subhypergraphs as the explanation. Four synthetic datasets were introduced in this paper, although none of them are open-sourced.

**Strengths:**

1. This paper is the second work on hyperGNN explainability.
2. They introduce a new benchmark that contains four synthetic datasets to evaluate hyperGNN explanations.

**Weaknesses:**

1. Related work is not thoroughly reviewed, and more recent studies should be discussed. The limitations of existing approaches for GNN explainability in the context of hyperGNNs are not clearly addressed. Since most existing methods are model-agnostic, they should be able to provide explanations for hyperGNNs too.
2. The newly introduced datasets are not provided.
3. The presentation is not very good. For example, the caption of Fig 2 is confusing.
4. The performance improvements on real world datasets seems minimal.
5. The evaluation on the global-level explainability is too few. It doesn't seem that this explainer actually produce global-level explanations.
6. Some concerns about motivation and metrics. See questions.

**Questions:**

1. In Line 171, it says "our goal is to produce an explanation subhypergraph" for the local explainer. However, a single subhypergraph may not be enough to fully explain the prediction. Why do you only consider to produce a single explanation subhypergraph for each graph sample?
2. The proposed method doesn't look to be specifically designed for hyperGNNs. For example, modules like Gumbel-Softmax samplers and global clustering. Why don't apply your method also to traditional GNNs? And why not use the explainers for those traditional GNNs, (which didn't use Gumbel-Softmax samplers, but some other methods to pick crucial subgraphs) to explain hyperGNNs?
3. For the Fidelity evaluation, why do you only evaluate the Fidelity-, how about Fidelity+?

---

> ### Author Response · Authors · 2024-11-23
>
> We thank the reviewer for their time in leaving such constructive feedback. We would like to address this below.
>
> _Related work is not thoroughly reviewed, and more recent studies should be discussed. The limitations of existing approaches for GNN explainability in the context of hyperGNNs are not clearly addressed. Since most existing methods are model-agnostic, they should be able to provide explanations for hyperGNNs too._
>
> Due to the space constraints we only provided a brief summary of the related works, to create a general idea of the field. If the reviewer has a concrete list of studies or related work that they think it is suitable to include, we are happy to take these suggestions into account.
>
> Regarding existing GNN methods, please refer to our overall comment.
>
> _The newly introduced datasets are not provided._
>
> In our supplementary material, we included the code for our model, together with the code to generate the synthetic datasets. Upon acceptance, we plan to release publicly both the code and the generated synthetic datasets.
>
> _The presentation is not very good. For example, the caption of Fig 2 is confusing._
>
> We are sorry for the potential clarity issues in our work. Section 5.1 Dataset Construction provides helpful context. Figure 2.a and 2.b shows samples of the “base hypergraph” used as the backbone for our datasets. Figure 2.c and 2.d depicts the motifs attached to the  base hypergraph to create the dataset. Finally, Figure 2.e depicts a small example of a dataset, containing a random hypergraph as a base, and 2 house-like motifs attached to the base. We would take all the suggestions from the reviewers into account to improve the clarity of the presentation. If the reviewer has any additional questions which might impact the understanding of the method we are happy to address them.
>
> _The performance improvements on real world datasets seems minimal._
>
> We would like to push back on the reviewer’s interpretation and argue that our results consistently demonstrate SHypX’s strong outperformance. We acknowledge that on real-world datasets, even simple baselines like gradient do well on the Fid-Acc metric; this motivated our introduction of the generalized fidelity metrics to better discern the differences in explainer quality. There we see significant gains of SHypX. We also highlight on these real datasets, SHypX explanations are smaller than the baselines’, so it is also superior in terms of concision.
>
> Finally, we would like to reiterate that SHypX produces the most faithful and concise explanations across every dataset we tested. In the results of Table 1 and 2, each number is averaged over ~1000 nodes or all of the nodes in the hypergraph, and in Figure 3, we study the performance of all methods across a range of explanation budgets – SHypX’s outperformance is consistent and robust, with its curve clearly beating baselines and demonstrating a smooth faithfulness-concision tradeoff.
>
> _The evaluation on the global-level explainability is too few. It doesn't seem that this explainer actually produce global-level explanations._
>
> We provide a series of qualitative results presenting the global concepts discovered by our model. The discovered sub-hypergraphs reflect the label semantics.
>
> For example, in Figure 4, for class 0 which correspond to the nodes that are part of the random hypergraphs (the “base” of the dataset to which we attach the motifs),  all the concepts are various random sub-hypergraphs frequently seen in the  dataset. Class 1 represents the top of the house, and the concept discovered by our model is exactly a house with the target node on the top. Class 2 represents the middle of the house so the global explanation capture a house with the target node one of the nodes connecting the roof to the main body of the house, while Class 3 represents the bottom of the house so the global explanation discovered by our house extract a fragment of the house, with the target node being one of the nodes that are not part of the “roof”. Additional visualizations are in the supplementary material in Figure 5, Figure 6 and Figure 7.
>
> In addition to these visual evaluations, we also provided the concept completeness score in Table 5 of the Supplementary Material. For all the datasets, our model obtained a high score, demonstrating that the explainer produces accurate global-level explanations. (Concept completeness was introduced in prior work. It quantifies how well we do on the original task of the hyperGNN if the original (i.e. node) features are replaced by the concept label only.)
>
> CONTINUED

---

> > ### Author Response · Authors · 2024-11-23
> >
> > _In Line 171, it says "our goal is to produce an explanation subhypergraph" for the local explainer. However, a single subhypergraph may not be enough to fully explain the prediction. Why do you only consider to produce a single explanation subhypergraph for each graph sample?_
> >
> > We are not entirely sure what the reviewer refers to by “a single subhypergraph may not be enough to fully explain the prediction”. For one node, disconnected components detached from the node can’t influence the prediction associated with that node, since the message passing will not receive the information from those disconnected nodes. Thus, the explanation can only be a single subhypergraph attached to the node. We also note that for a global explanation that holds across multiple instances or a full class, one may indeed obtain multiple subgraph explanations that represent a common concept or a class.
> >
> > _The proposed method doesn't look to be specifically designed for hyperGNNs._
> >
> > You are correct that the techniques we leverage here may be applied to traditional GNNs. However, we choose to focus on the hypergraph problem which is relatively neglected and also more challenging. As for existing GNN methods, please see the overall comment. We show that GNNexplainer struggles and suggest why SHypX has an advantage.
> >
> > _For the Fidelity evaluation, why do you only evaluate the Fidelity-, how about Fidelity+?_
> >
> > In Appendix E we provide an extended discussion on why Fid+ is not an appropriate metric to assess the concision, especially in the hypergraph world. This is mainly due to the large amount of information that the complement of a subhypergraph still preserves (for a better understanding, we kindly ask the reviewer to read the entire section together with the visual depiction). Instead, we are measuring the concision using the size and density metrics, which we show to be a much more accurate metric for capturing the minimality of the explanation.

---

> ### Comment · Area_Chair_fv1T · 2024-11-25
> **[URGENT]: react to the authors' rebuttal**
>
> Dear Reviewer,
>
> The authors have posted a rebuttal to the concerns raised. Please read the rebuttal and see if this changes your opinion on the work. We have only two days left. Hence, we urgently need your reactions to the rebuttal.
>
> best,
>
> AC

---

> ### Comment · Reviewer_agpw · 2024-12-02
>
> Thanks a lot for the author's rebuttal. My concerns are partially resolved. However, I agree with other reviewer's opinion that the motivation of proposing explainers for HyperGNNs stays unclear. Since GNNExplainer can be utilized to explain HyperGNNs, other GNN explainers should be compared as the baselines too. GNNExplainer is a relatively poor baseline in this domain. Therefore, I would maintain the score. Thanks a lot for your rebuttal. I appreciate it.

---

### Official Review · Reviewer_KdSZ · 2024-10-30

**Soundness:** 3
**Presentation:** 3
**Contribution:** 2
**Rating:** 5
**Confidence:** 5

**Summary:**

This paper addresses the post-hoc explainability of hypergraph neural networks (HyperGNNs) for the task of node classification in hypergraphs. The authors propose a method for providing both instance-level and global-level explanations for these models. The core idea of their approach is to sample node-hyperedge pairs in the computation graph of a specific node in a way that minimizes the loss function. Additionally, they introduce a set of synthetic hypergraphs to evaluate their method. Their experimental results demonstrate that the proposed method outperforms existing baselines.

**Strengths:**

- The paper is well-written and clearly presented.
- The authors address both instance-level and global-level explainability for HyperGNNs while latter has not been addressed previously.

**Weaknesses:**

- My main concern is that this paper only addresses explainability for the task of node classification in hypergraphs, which is a less interesting and important problem compared to (hyper)graph classification.
- The baselines used in this paper are not sufficient. While there might not be many related works on explainability in hypergraphs, explainability methods for graph neural networks could be applied and compared with the proposed method.
- The quantitative evaluation on real-world datasets shows very low performance, and the differences between the various baselines are minimal, making it difficult to determine if the proposed method performs significantly better than the other baselines.
- While having synthetic datasets for an explainability method is crucial, it is not clear how the proposed synthetic datasets are particularly useful in this setting.

**Questions:**

- For the task of node classification, what would happen if a method returns only the first immediate neighbors of a node as the most important sub-hypergraph? Specifically, how would the fidelity metrics mentioned in this paper perform in this scenario? My main point is that having only an explainability model for node classification may not be of significant importance.

- Could you present a hypergraph as a bipartite graph and apply graph-based explainability models to compare their performance with your method? It would be particularly useful to try GNNExplainer, as this method is very intuitive, and it is not immediately clear why it would not perform well in your setting.

- For the global explainer, you are introducing yet another black-box, despite mentioning that SHypX "doesn’t rely on additional black-box networks." Could you clarify this contradiction?

- It is necessary to visualize the results of the explanations for at least the synthetic datasets to visually assess which sub-hypergraphs SHypX identifies as important.

- Figure 4 is very confusing, and the results you are trying to convey are not clear. Is there an alternative way to evaluate global explanations that might be more effective?

---

> ### Author Response · Authors · 2024-11-23
>
> We warmly thank the reviewer for their time and questions! We are glad that you appreciate that we are the first to address both local and global explanations for hyperGNNs and regard our manuscript as well-written. We address your points below:
>
> _My main concern is that this paper only addresses explainability for the task of node classification in hypergraphs, which is a less interesting and important problem compared to (hyper)graph classification._
>
> We thank the reviewer for their comments. Firstly, we would like to note that one cannot discern node classification more important than hypergraph classification, as it boils down to the task at hand. Many relevant problems, e.g. predictions about an entity in a complex network, are formulated as node classification. Secondly, the techniques we develop and test on node classification can be easily adapted to hypergraph classification as well as regression tasks; we briefly discuss these possibilities in Appendix A. However, we are not aware of any real-world benchmark for hypergraph-level classification. This is largely because hypergraph representation learning is a relatively new and evolving area of research. However, we would be glad to extend our experiments to include hypergraph-level tasks if the reviewer could kindly suggest relevant datasets.
>
> _The baselines used in this paper are not sufficient. While there might not be many related works on explainability in hypergraphs, explainability methods for graph neural networks could be applied and compared with the proposed method._
>
> Please refer to our overall comment.
>
> _The quantitative evaluation on real-world datasets shows very low performance, and the differences between the various baselines are minimal, making it difficult to determine if the proposed method performs significantly better than the other baselines._
>
> We would like to push back on the reviewer’s interpretation and argue that our results consistently demonstrate SHypX’s strong outperformance. We acknowledge that on real-world datasets, even simple baselines like gradient do well on the Fid-Acc metric; this motivated our introduction of the generalized fidelity metrics to better discern the differences in explainer quality. There we see significant gains of SHypX. We also highlight on these real datasets, SHypX explanations are smaller than the baselines’, so it is also superior in terms of concision.
>
> Finally, we would like to reiterate that SHypX produces the most faithful and concise explanations across every dataset we tested. In the results of Table 1 and 2, each number is averaged over ~1000 nodes or all of the nodes in the hypergraph, and in Figure 3, we study the performance of all methods across a range of explanation budgets – SHypX’s outperformance is consistent and robust, with its curve clearly beating baselines and demonstrating a smooth faithfulness-concision tradeoff.
>
> _While having synthetic datasets for an explainability method is crucial, it is not clear how the proposed synthetic datasets are particularly useful in this setting._
>
> Our primary motivation in introducing the synthetic datasets is to produce a challenging and discerning evaluation for hyperGNN explainers (see Section 5.1). Concretely, in creating the datasets, we ensured that the label strongly depends on the structural connectivity of the hypergraph.  This way, we make sure that the hypergraph model that we are explaining are models that are forced to actually capture higher-order connectivity. Moreover, we would like to point out that these datasets represent a higher-order version of the pairwise datasets used in the graph explainability community. These datasets constitute a challenging testbed and are widely used for testing graph explainers [1,2,3].  If the reviewer disagrees with some specific reasoning contained there, we would appreciate a more specific pointer.
>
> [1] Ying, Zhitao, et al. "Gnnexplainer: Generating explanations for graph neural networks." Advances in neural information processing systems 32 (2019).
>
> [2] Vu, Minh, and My T. Thai. "Pgm-explainer: Probabilistic graphical model explanations for graph neural networks." Advances in neural information processing systems 33 (2020): 12225-12235.
>
> [3] Yuan, Hao, et al. "On explainability of graph neural networks via subgraph explorations." International conference on machine learning. PMLR, 2021.
>
> [4] Magister, Lucie Charlotte, et al. "Gcexplainer: Human-in-the-loop concept-based explanations for graph neural networks." arXiv preprint arXiv:2107.11889 (2021).
>
> [5] Magister, Lucie Charlotte, et al. "Concept distillation in graph neural networks." World Conference on Explainable Artificial Intelligence. Cham: Springer Nature Switzerland, 2023.
>
> CONTINUED

---

> > ### Author Response · Authors · 2024-11-23
> >
> > _For the task of node classification, what would happen if a method returns only the first immediate neighbors of a node as the most important sub-hypergraph?_
> >
> > Returning the immediate neighbors as an explanation would in general perform poorly. We consider H-RandHouse and Zoo as examples. In H-RandHouse, the motifs are greater than a 1-hop neighborhood and it’s not possible to reproduce the model’s high accuracy this way. In Zoo, our hyperGNN has just 1-layer, so taking the immediate neighbors trivially achieves perfect fidelity in this case; however, these explanations fails on the concision criterion since they comprise the entire computational subgraph. Furthermore, returning the immediate neighbors does not allow the user to trade off the size vs “quality” (i.e. faithfulness) of the explanation, which SHypX can do (Figure 3).
> >
> > | | | fid-acc | fid-kl | fid-tv | fid-xent | size | density  |
> > |---|---|---|---|---|---|---|---|
> > |
> > | H-RandHouse | first immediate neighbors | 0.50 | 1.08	| 0.44 | 1.63 | 6.9 | 0.16 |
> > | Zoo | first immediate neighbors | 0.00 | 0.00	| 0.00 | 0.18 | 991 | 1.00 |
> >
> >
> > _Could you present a hypergraph as a bipartite graph and apply graph-based explainability models to compare their performance with your method? It would be particularly useful to try GNNExplainer, as this method is very intuitive, and it is not immediately clear why it would not perform well in your setting._
> >
> > Treating hypergraphs as bipartite graphs and applying standard graph techniques to process them is a parallel direction of study in hypergraph representation learning that researchers are testing as an alternative to building dedicated hypergraph tools [1,2]. However, the results we are having so far from these attempts show that, treating both nodes and hyperedges as similar nodes, negatively impact the performance. The hypergraph-dedicated message passing generally outperforms the bipartite approach [4,5]. Moreover, even experimenting with heterogeneous graph architectures that should be able to distinguish between the 2 types of entities still underperform compared to the more powerful hypergraph networks. This indicates that naively treating the hypergraph as a bipartite graph is not a recipe for success. However, analysing this in the context of explainability is an interesting area of research which deserve a paper on its own. In addition to this, we are not aware of any explainability techniques dedicated to heterogeneous methods, which can bring additional challenges to the task.
> >
> > If the reviewer is wondering how a GNNexplainer adapted for hypergraphs would perform, we refer them to our overall comment.
> >
> > [1] Huang&Yang. Unignn: a unified framework for graph and hypergraph neural networks.
> >
> > [2] Yang et al. Semi-supervised Hypergraph Node Classification on Hypergraph Line Expansion
> >
> > [3] Wang et al. Heterogeneous graph attention network.
> >
> > [4] Chien et al. You are AllSet: A Multiset Function Framework for Hypergraph Neural Networks
> >
> > [5] Wang et al. Equivariant Hypergraph Diffusion Neural Operators
> >
> > _For the global explainer, you are introducing yet another black-box, despite mentioning that SHypX "doesn’t rely on additional black-box networks." Could you clarify this contradiction?_
> >
> > Thank you for the question. There is no black-box here. The global explanation relies on a k-means clustering of the latent representation to identify the concepts, followed by majority voting to associate each concept with a class. Both k-means and majority vote are generally considered interpretable. These concepts are visualized using the instance-level explainer, which directly optimizes for the explanation subhypergraph without introducing any black box model. If the reviewer is asking about a different element of the pipeline, we would be happy to clarify.
> >
> > _It is necessary to visualize the results of the explanations for at least the synthetic datasets to visually assess which sub-hypergraphs SHypX identifies as important._
> >
> > We have prepared several more examples as requested. We would also like to note that the current paper contains visualizations of the important concepts discovered by our model both for the synthetics and real-world datasets.  For global-level explanations these can be found in Figure 4 of the main paper, together with Figure 5,  Figure 6,  Figure 7 in the Supplementary material. For the instance-level explanation we provide them in Figure 8 and Figure 9 in the Supplementary Material.
> >
> > CONTINUED

---

> > > ### Author Response · Authors · 2024-11-23
> > >
> > > _Figure 4 is very confusing, and the results you are trying to convey are not clear. Is there an alternative way to evaluate global explanations that might be more effective?_
> > >
> > > In Figure 4 we are showing the results for the global-level explanations for H-RandHouse obtained when using 10 concepts. Each sub-hypergraph is the representative of one concept, and the labels identify the class associated with that concept.
> > >
> > > We note that for class 0 which corresponds to the nodes that are part of the random hypergraphs (the “base” of the dataset to which we attach the motifs),  all the concepts are various random sub-hypergraphs frequently seen in the dataset. Class 1 represents the top of the house, and the concept discovered by our model is exactly a house with the target node on the top. Class 2 represents the middle of the house so the global explanation capture a house with the target node one of the nodes connecting the roof to the main body of the house, while Class 3 represents the bottom of the house so the global explanation discovered by our house extract a fragment of the house, with the target node being one of the nodes that are not part of the “roof”. This suggests that the discovered sub-hypergraphs actually match the semantic of the labels.
> > >
> > > In addition to these visual evaluations, we also provided the concept completeness score in Table 5 of the Supplementary Material. For all the datasets, our model obtained a high score, demonstrating that the explainer produces accurate global-level explanations.

---

> > > > ### Comment · Reviewer_KdSZ · 2024-11-23
> > > > **Thank you for your response.**
> > > >
> > > > Thank you for your response. My main concern remains around the use of GNNExplainer.
> > > >
> > > > I still believe the best approach to using GNN explainers for hypergraphs is to treat them as bipartite graphs. While the authors have shown results for GNNExplainer, they convert the hypergraph to a graph, which is a lossy approach and could indeed result in poor outcomes.
> > > >
> > > > Furthermore, I would like a more thorough discussion on why it is necessary to have an explainability approach specifically for hypergraph neural networks and why GNN-based approaches (with minimal changes) are not sufficient. I am not convinced by the authors' discussion so far.
> > > >
> > > > Additionally, it seems like SHypX performs well on Fid -, but not as well on Fid+. This discrepancy needs further investigation. I would also like to see the results for Fid+ using GNNExplainer for a more comprehensive comparison.

---

> > > > > ### Author Response · Authors · 2024-11-23
> > > > >
> > > > > Thank you for your quick reply!
> > > > >
> > > > > Concerning GNNexplainer: we are not converting the hypergraph to a graph, so there is no lossinness here. We are allowing GNNexplainer to learn over the hyperedge index of the hypergraph rather than the edge index of the graph. Besides this, we are not sure what the reviewer means by treating the hypergraph as a bipartite graph -- the hyperGNN architectures that we want to explain take hypergraphs as inputs and derive their advantage from treating the input as a hypergraph rather than a bipartite graph. Doing the bipartite graph explanation problem would not allow us to explain hyperGNNs.
> > > > >
> > > > > Concerning Fid+: In Appendix E we provide an extended discussion on why Fid+ is not an appropriate metric to assess the concision, especially in the hypergraph world. This is mainly due to the large amount of information that the complement of a subhypergraph still preserves (for a better understanding, we kindly ask the reviewer to read the entire section together with the visual depiction). Instead, we are measuring the concision using the size and density metrics, which we show to be a much more accurate metric for capturing the minimality of the explanation. If the reviewer disagrees with the reasoning there, we welcome a more detailed discussion.

---

> > > > > > ### Comment · Reviewer_KdSZ · 2024-11-26
> > > > > >
> > > > > > **We adapt PyG’s GNNexplainer implementation for hypergraphs and otherwise use defaults, thresholding edge weights**
> > > > > >
> > > > > > Regarding GNNExplainer, I believe that the adaptation you mentioned might be contributing to the poor results observed. I am particularly interested in seeing the results when you convert the hypergraph to a bipartite graph. In this representation, you would have one set of nodes representing the original nodes and another set representing the hyperedges, with the adjacency matrix constructed accordingly. This approach would allow you to maintain the use of a hypergraph for your hyperGNN while providing GNNExplainer with a bipartite representation.
> > > > > >
> > > > > > Additionally, I noticed that you reported a fid-acc of 0.00 and a fid-kl of almost 0.00 for most datasets in your comparison with GNNExplainer. These results are quite difficult to interpret. Although you mentioned that Fid+ is not an appropriate metric, including it might still offer some insights.

---

> > > > > > > ### Author Response · Authors · 2024-12-04
> > > > > > >
> > > > > > > Our adaptation of GNNexplainer follows what you are describing. The hyperedge index contains pairs where each pair references one hypernode and one hyperedge (compared to the original GNNexplainer operating over the graph edge index where each pair references two nodes joined by an edge). GNNexplainer can "choose" which of these links it deems important, so this is isomorphic to the bipartite setup.
> > > > > > >
> > > > > > > We are not sure what the reviewer means by a fid-acc and fid-kl of almost 0.00 being hard to interpret. Note that the entries listed as 0.00 are true 0s (not rounded). This indicates that the distribution of predictions produced by SHypX explanations is extremely close to the distribution of original predictions over the input hypergraph. Evaluating the complement (via Fid+) is motivated by the desire to have explanations that are "necessary" i.e. not needlessly verbose. [1] This is more simply and intuitively measured by size and concision, so we advocate relying on those metrics instead. We do report some Fid+ metrics in the appendix Table 6 and indeed find it much harder to interpret, with the less faithful explanations sometimes scoring well. In some cases we also observe that concise explanations leading to large complements, which is desirable, but punitive for Fid+.
> > > > > > >
> > > > > > > [1] Amara, Kenza, et al. "Graphframex: Towards systematic evaluation of explainability methods for graph neural networks." arXiv preprint arXiv:2206.09677 (2022).

---

### Official Review · Reviewer_4e64 · 2024-10-31

**Soundness:** 3
**Presentation:** 3
**Contribution:** 4
**Rating:** 5
**Confidence:** 4

**Summary:**

The paper introduces SHypX, the first model-agnostic post-hoc explainer for hyperGNNs that provides both local and global explanations.
At the local level, it finds salient subhypergraphs to explain individual predictions using Gumbel-Softmax sampling, while balancing faithfulness and concision.
For global explanations, it extracts concepts by clustering network representations and visualizing representative examples.

In experiments, the paper introduces four synthetic hypergraph datasets and generalized fidelity metrics for proper evaluation.
Through experiments on both synthetic and real datasets, SHypX demonstrates superior performance over existing baselines while maintaining architecture independence, making it a significant contribution to hypergraph machine learning interpretability.

**Strengths:**

The paper presents multiple novel contributions:
1. The first to provide both local and global explanations for hyperGNNs.
2. Introduces a new sampling-based approach for local explanations that avoids using attention mechanisms.
3. It develops techniques specifically designed for hypergraph structures.
4. The synthetic datasets also represent an original contribution by creating structure-dependent tasks for evaluating hypergraph explainability.
5. This work addresses an important gap in making hyperGNNs more interpretable.

**Weaknesses:**

**Insufficient Analysis of Graph-to-Hypergraph Explainability**

**Constrained Real-World Evaluation**

**Questions:**

**1.Insufficient Analysis of Graph-to-Hypergraph Explainability**
The paper provides some reasons why hypergraphs need specialized explainers, such as larger search spaces and structural differences. However, it lacks an in-depth analysis of what specifically would fail if traditional GNN explainers were used on constructed hypergraphs, which can be represented as graphs. Including a comparative study that illustrates concrete failure cases when applying regular GNN explainers to hypergraphs would significantly strengthen the motivation for developing specialized hypergraph explainers.

**2. Constrained Real-World Evaluation**
- The real-world datasets used in the paper are relatively small, which limits the ability to generalize the results.
- Moreover, the paper acknowledges that the selected datasets may not sufficiently test the model's ability to understand hypergraph structure, given that even simple MLPs achieve competitive performance. While the inclusion of synthetic datasets partially addresses this limitation, it would be more compelling to include more complex real-world hypergraph datasets that present a greater challenge. Such datasets would better demonstrate the practical utility and robustness of the proposed method. - Furthermore, the constrained real-world evaluation raises the question: does the limited complexity and availability of these datasets indicate that the problem lacks substantial practical applications?

Minor Comment: In Figure 1, what does the "concept-to-class decision tree" icon represent? Providing a brief explanation would help improve clarity.

---

> ### Author Response · Authors · 2024-11-23
>
> We warmly thank the reviewer for their thoughtful comments, for valuing our work as a significant contribution to hypergraph machine learning explainability, and for appreciating our multiple novel contributions in tackling both local and global explanations, sampling for hypergraphs, and synthetic datasets, among others.
>
> We address the concerns below:
>
> _Insufficient Analysis of Graph-to-Hypergraph Explainability_
>
> We fully acknowledge there are similarities between the hyperGNN and GNN variants of the problem; this is why we have taken inspiration from the GNN methods. Our contribution is to design an explainer for hyperGNNs that is simple and effective rather than to closely adapt existing GNN explainers and identify where they fail. We believe the GNN-to-hyperGNN challenge comes down to the larger search space for potential explanation subhypergraphs, as noted by the reviewer. How this might manifest as failure modes of existing GNN explainers could vary greatly, but we summarize a few high-level themes:
> * Some methods, such as GNNExplainer, rely on learning a fractionally-relaxed adjacency matrix and obtain the actual subgraph by thresholding. While this seems to have sufficed for GNNs, the performance is notably worse for hyperGNNs. We do a case study with GNNexplainer on our benchmarks and discuss the challenges – please refer to our overall comment.
> * Many methods directly explore the combinatorial space of subgraphs e.g. SubgraphX by MCTS or XGNN by generating a graph with policy gradients. These suffer from the large action space of the hypergraph compared to the graph, as we have discussed (line 194).
> * Concept-based methods rely on a clusterization of the latent space. While this was empirically validated on graph-based methods, we are the first work that validates a similar behaviour on hypergraphs.
> * Global concept-based explainers designed for graphs (such as GCExplainer and CDM) are relying on n-hop neighborhood to provide the explanation. While for graphs this still outputs a reasonable-sparse structure, in the case of the higher-order structure, sub-hypergraphs obtained by extracting n-hop neighbourhoods are hard to interpret. We provide a visual comparison of this phenomena in Figure 8 where the top row displays explanations extracted with k-hop neighbourhoods (already very dense even for a small k, k=3), while the bottom row displays our concise explanations.
>
> Finally, please refer to our overall comment for a case study that illustrates a failure case of applying a GNN explainer to hyperGNN.
>
>
> _Constrained Real-World Evaluation_
>
> To test our model in real-world scenarios, we are using the established benchmarks used in the hypergraph representation learning community [1-4]. We select a subset of datasets where using the hypergraph structure produces a substantial benefit, such that we can fairly test the capability of our structure-based explainer. While it is beyond the scope of the current paper, we are keen to test our model in more challenging scenarios such as datasets with higher number of nodes or more complex interactions. We welcome any suggestion of such benchmarks if the reviewer has concrete examples. Moreover, moving from the synthetic setup to the real-world domain is still a challenge in the graph community as well, even if the domain is much older and well developed compared to the hypergraph one, with most of the current advances in graph explainability being validated on the synthetic BA-based datasets [5-6] . Please remember that this is an explainability work and thus synthetic datasets are of utmost importance.
>
> We also highlight that the lack of benchmarks is a well-known issue in the hypergraph community. This is clearly not a problem related to a potential “lack of substantial practical applications”, as several works [7-9] suggested the importance of higher-order interactions in modelling relations from various domains. However, hypergraph representation learning is a much more novel and slow-paced domain compared to its graph counterpart (with the first popular architecture being published in 2019 compared to GCN which started to be popular in 2017) so we can not expect the same level of resources and data available as in the graph literature.  In fact, GNN literature evolved around the standard Cora-Citeseer benchmark for several years until the current large-scale datasets were made available.
>
> CONTINUED

---

> > ### Author Response · Authors · 2024-11-23
> >
> > [1] Feng et al. Hypergraph Neural Networks
> >
> > [2] Chien et al. You are AllSet: A Multiset Function Framework for Hypergraph Neural Networks
> >
> > [3] Wang et al. Equivariant Hypergraph Diffusion Neural Operators
> >
> > [4] Duta et al. Sheaf Hypergraph Networks
> >
> > [5] Magister, Lucie Charlotte, et al. "Concept distillation in graph neural networks." World Conference on Explainable Artificial Intelligence. Cham: Springer Nature Switzerland, 2023.
> >
> > [6] Yuan, Hao, et al. "On explainability of graph neural networks via subgraph explorations." International conference on machine learning. PMLR, 2021.
> >
> > [7] Murgas et al. Hypergraph geometry reflects higher-order dynamics in protein interaction networks
> >
> > [8] Gatica et al. High-order interdependencies in the aging brain
> >
> > [9] Jost et al. Hypergraph Laplace operators for chemical reaction networks
> >
> >
> > _In Figure 1, what does the "concept-to-class decision tree" icon represent?_
> >
> > We apologize for the confusion and appreciate the correction; that icon should have been updated to the “MajorityVote” function which appears in Eq. 6. This will be fixed.

---

> > > ### Comment · Reviewer_4e64 · 2024-11-23
> > >
> > > Thank you for addressing my comments.
> > >
> > > After reviewing your responses, I still have the following concerns, which I believe other reviewers might share as well:
> > >
> > > **W1.**
> > >
> > > 1. In the current submission, the discussion on this critical limitation is insufficient. This is not a critique of your work but rather a suggestion to strengthen its framing. I understand and appreciate your goal of proposing a simple yet effective hyperGNN explainer. However, a clearer and more detailed discussion of this limitation would significantly enhance the motivation and impact of your work.
> > > 2. While I acknowledge the constraints of the rebuttal phase, including the limited time for extensive experiments, I find that a single comparison experiment with GNNExplainer is not convincing enough. Meanwhile, I expect more evidence to support your statements, such as space explore explainers are computationally in-feasible for hypergraphs. I strongly encourage the authors to consider revising and resubmitting to a future venue. While I do not doubt the contribution of your work, the current submission feels incomplete from my perspective.
> > >
> > > **W2.**
> > >
> > > I appreciate the acknowledgment of real-world evaluation constraints in your response. While I agree that constructing evaluation datasets is not the primary objective of this paper, I strongly encourage the authors to consider contributing to benchmark construction in the future if this remains a focus of their research. Without a solid and widely accepted benchmark, it will be challenging for the community to evaluate and validate your conclusions. Additionally, if the construction of real-world datasets is indeed this challenging, it raises a question about the prevalence of hypergraphs in real-world applications. A brief discussion on this point in the submission could be valuable.
> > >
> > > To summarize, W1 remains my primary concern. While I appreciate the efforts in addressing the points raised, I will not be changing my score.

---

### Official Review · Reviewer_NGZs · 2024-11-03

**Soundness:** 2
**Presentation:** 2
**Contribution:** 2
**Rating:** 3
**Confidence:** 4

**Summary:**

The paper studies a relatively unexplored problem of hypergraphs neural networks explanation for the task of node classification, both from the local and global perspective. The authors describe two methods they use for local and global explanations. They
then benchmark the methods on synthetic and real-world datasets against current baselines, notably with an updated fidelity metric.

**Strengths:**

- The topic is interesting, and still in unexplored territory.
- The paper is well written, the experiments seem to yield good results and the experimental design looks solid.

**Weaknesses:**

- There is not a lot of theory in this paper, and the methods used are pretty straightforward.
- The literature could be a bit more developed (the one other baseline on the problem, global explanations of GNNs, mean field approximation in the hypergraphs context).
- The local and global explanations techniques are completely decoupled, with the global
explanation in particular being transposed from a method already studied for GNNs. This is a liitle surprising, as the titled implied a deeper relationship.
- There should be more local and global explanations derived from this work and derived
from the GNNs explanation literature: benchmark different global explainer heuristic
based on the local explanation (as opposed to just one); for the local explanation, an
approximation technique with a more refined method, or at least explain why other
methods would not work.

Overall the paper needs a major revision.

**Questions:**

- Line 212 you use a mean-field approximation, can you justify this more, add some
literature that justifies such approximations for hypergraphs? Why can you do that
(specially in the context of hypergraphs)? Are there other approximations you explored
or considered? This is a pretty strong assumption.

- Please define a concept line 251; it seems to be specific to a GNN, and to be a cluster
of points in the GNN embedding space, and its ”representative” is the node closest to
the geometric center in the embedding space, is that correct?

- Line 249: Can you specify what the latent embedding space is for a node? Is it the
final layer’s output before the softmax?

- The local and global explainer seem to be two completely decoupled methods; please
point this out in the paper. How are they related?

- Have you tried other types of global explanations (e.g., GLGex-
plainer [1]) based on this local explainer?

- Have you tried other local explanations techniques (there are plenty of work [2])?

- Has a similar sampling method for local explanations been used in GNN explanations?

Some minor comments:

– Equation (3) line 209 is not clear, I would write it differently.

– Some typos: “explainiability” line 75, “P r(v ∈ esub = 0)” line 206

– “coherent explanations” line 80: please clarify what this means.

– What is “InfoNCE” line 106? add context about noise-contrastive estimation.

– Line 259, GCExplainer is mentioned in the method, but there has not been a
reference since the introduction. It should be added on line 244 along with the
Magister paper.

– For the results, it would be helpful to separate the local from the global explanations.

– The paragraph starting line 259 compares GCExplainer with the method of the
paper, this is not clear, as it is never explained what GCExplainer does.

[1] Azzolin, Steve, Antonio Longa, Pietro Barbiero, Pietro Liò, and Andrea Passerini. "Global explainability of gnns via logic combination of learned concepts." arXiv preprint arXiv:2210.07147 (2022).

[2] Kakkad, Jaykumar, Jaspal Jannu, Kartik Sharma, Charu Aggarwal, and Sourav Medya. "A survey on explainability of graph neural networks." arXiv preprint arXiv:2306.01958 (2023).

---

> ### Author Response · Authors · 2024-11-23
>
> We thank the reviewer for their time and thoughtful comments. We are glad that you appreciate the value of the topic, the presentation of our paper, our strong results, and experimental design! We address each of the reviewer’s points below.
>
> _There is not a lot of theory in this paper, and the methods used are pretty straightforward._
>
> As the reviewer rightly highlighted, designing explainers for hypegraph models is an unexplored territory, which we believe limits the applicability of hypergraphs, particularly in sensitive domains. Our proposed method is indeed straightforward, making it compatible with any existing hypergraph approach without introducing significant computational overhead, an important feature for explainer architectures. While this is not a theoretical paper, the components of the proposed method are clearly motivated and it generates high-quality explanations, as demonstrated in our experiments. This is well-aligned with the requirements of ICLR conference, which welcome not only theoretical papers, but also novel empirical work.
>
> _The literature could be a bit more developed..._
>
> Due to the space constraints we only provided a brief summary of the related works, to create a general idea of the field. Following the reviewer's suggestion, we will include a more detailed literature review in the supplementary material together with more details on preliminary methods. If the reviewer has any particular questions regarding the baselines or the method used in our model we are happy to answer them.
>
> _The local and global explanations techniques are completely decoupled..._
>
> We draw attention to the fact that the two explainers are not decoupled. As explained in Section 4.2 and in Figure 1, the local explainer may be used as a stand-alone explainer in order to produce instance-level explanations; furthermore, the global explainer relies on the local explainer to produce concise and legible explanation subhypergraphs based on the extracted concepts (line 256). This is a novelty to the global GNN method the reviewer alludes to, and without, the global explanations (e.g. Figure 4) would be dense 3-hop neighborhoods with poor concision.
>
> _There should be more local and global explanations derived from this work and derived from the GNNs explanation literature..._
>
> Please refer to our overall comment.
>
> _you use a mean-field approximation, can you justify this more..._
>
> We fully acknowledge that the mean field approximation is a simplifying assumption. Though stringent, it is frequently used in variational inference to make progress on an intractable joint probability. While we have not seen this explicitly in the hypergraph literature, the complex probability distribution we are dealing with fits the mould of those variational problems; furthermore, the approximation underlies (graph) methods like GNNexplainer which independently estimates for each edge whether that edge should exist in the explanation. Finally, we believe the validity of the approximation is justified by our strong results. Even though we use an approximation to optimize for the explanation subhypergraph, those explanations successfully replicate the original predictions, as reflected in the fidelity metrics.
>
> _Please define a concept line 251..._
>
> Thank you for the suggestion, we will clarify this in the final version. In our work, ‘concept’ denotes units of information that are easy to understand and interpret by humans, and frequently appear in the analysed data, following Ghorbani et al. (2019). Methods using concept-based explanations are popular in the explainability literature for vision (where image patches can be identified as concepts), language (where certain phrases or a bag of words can be identified as concepts) and also graph data (where network subgraphs can be interpreted as concepts). In explainability, a concept constitutes a repetitive substructure of the original input modality found across multiple input examples. On a technical level, concepts in neural networks can be discovered via clusters in the latent space. The clustered embeddings often exhibit similar semantics. For example, in the case of (hyper)GNNs, nodes clustered together have been found to exhibit similar feature vectors and neighbourhoods. Following Magister et al. (2021), we extract the nodes closest to the geometric centre of a cluster as the most representative candidates for the concept.
>
> _Can you specify what the latent embedding space is for a node?_
>
> Thank you for the question; we will make sure this is clearly stated in the final version. We are using the embedding after the final message passing layer. We expect similar concept extraction is possible using other late-stage layers in the architecture.
>
> _Have you tried other explanation..._
>
> Please refer to our overall comment.
>
> CONTINUED

---

> > ### Author Response · Authors · 2024-11-23
> >
> > _Has a similar sampling method for local explanations been used in GNN explanations?_
> >
> > There are some fundamental differences between sampling subhypergraphs and subgraphs. Whereas for graphs, it is natural to consider whether an edge exists between pairs of nodes, for hypergraphs, this is lifted to the problem of node membership in a hyperedge. Our notion of a subhypergraph is similar to HyperEX’s, though the manner in which we obtain that subhypergraph is totally different. As for the use of the Gumbel-Softmax, this has been proposed for feature selection [1] and to sample subgraphs for GNNs [2].
> >
> > [1] Chen, Jianbo, et al. "Learning to explain: An information-theoretic perspective on model interpretation." International conference on machine learning. PMLR, 2018.
> >
> > [2] M. S. Schlichtkrull, N. D. Cao, and I. Titov, “Interpreting graph neural networks for {nlp} with differentiable edge masking,” in International Conference on Learning Representations, 2021.

---

> > > ### Comment · Reviewer_NGZs · 2024-11-23
> > >
> > > Thanks for the rebuttal.
> > >
> > > My major concern (other reviewers also have the same) still remains regarding the comparisons with other existing methods. GNNExplainer is the first method in this space.
> > >
> > > I am unable to raise my score at this point. Another concern is the definition of concept itself. How much "human-interpretable" is it?

---

> > > > ### Author Response · Authors · 2024-11-24
> > > > **Thank you for your prompt reply!**
> > > >
> > > > Thank you for your prompt reply!
> > > >
> > > > We have chosen to compare with GNNExplainer, because it is the seminal work for GNN explainability and most papers use it as a baseline to compare to. We would also like to highlight that our work is one of the first two works on HGNN explainability and that our focus lies on demonstrating the ideas and working of our work, rather than adapting existing GNN methods, whose adaption would be a paper in itself.
> > > >
> > > > Regarding your concern for concept-based interpretability, we would like to highlight that this is a well established field. We provide a selection of papers (with many more existing) on concept-based explainability/interpretability below:
> > > > [1] Xuanyuan, Han, et al. "Global concept-based interpretability for graph neural networks via neuron analysis." Proceedings of the AAAI Conference on Artificial Intelligence. Vol. 37. No. 9. 2023.
> > > > [2] Azzolin, Steve, et al. "Global explainability of gnns via logic combination of learned concepts." arXiv preprint arXiv:2210.07147 (2022).
> > > > [3] Magister, Lucie Charlotte, et al. "Concept distillation in graph neural networks." World Conference on Explainable Artificial Intelligence. Cham: Springer Nature Switzerland, 2023.
> > > > [4] Bui, Tien-Cuong, and Wen-Syan Li. "Toward Interpretable Graph Classification via Concept-Focused Structural Correspondence." Pacific-Asia Conference on Knowledge Discovery and Data Mining. Singapore: Springer Nature Singapore, 2024.
> > > > [5] Kim, Sangwon, and Byoung Chul Ko. "Concept graph embedding models for enhanced accuracy and interpretability." Machine Learning: Science and Technology 5.3 (2024): 035042.
> > > > [6] Georgiev, Dobrik, et al. "Algorithmic concept-based explainable reasoning." Proceedings of the AAAI Conference on Artificial Intelligence. Vol. 36. No. 6. 2022.
> > > > [7] Yeh, Chih-Kuan, et al. "On completeness-aware concept-based explanations in deep neural networks." Advances in neural information processing systems 33 (2020): 20554-20565.
> > > >
> > > > To provide more of an intuition for concepts, we would like to also give the following explanation. Concept-based explanations are usually in the form of the input medium, for example, a subgraph for graph tasks or an image patch for image tasks. The explanations are human-interpretable because the user can usually reason about a selection of examples of the concept and the concept is a smaller part of the whole. We would also like to highlight the desirable properties defined for a concept established by Ghorbani et al. (2019) [8]: a concept should be meaningful, coherent and important, which we demonstrate in our paper.
> > > >
> > > > [8] Ghorbani, Amirata, et al. "Towards automatic concept-based explanations." Advances in neural information processing systems 32 (2019).
> > > >
> > > > Thank you for your time and consideration! Please let us know if this clears up your concerns.

---

### Author Response · Authors · 2024-11-23
**regarding GNN explainers**

We thank the reviewers for taking the time to comment so thoroughly on our paper! We would like to reply to a few common comments.

Several reviewers asked about comparisons to GNN methods and why explaining hyperGNNs is different from explaining GNNs.

First, we would like to note that porting methods from GNN to hyperGNNs often involves non-trivial adaptation. In the realm of (hyper)graph architectures, such developments have constituted several independent works in the literature (HGNN, HCHA, HNHN, HyperGCN etc.). Similarly, in the realm of explainability, the extension of computer vision explainers to effective GNN explainers have also themselves warranted single papers [1,2]. Our contribution is to design an explainer for hyperGNNs that is simple and effective rather than to closely adapt many existing GNN explainers and identify where they fail.

As a case study, we investigate GNNexplainer as a baseline, and this illustrates some new challenges of the hyperGNN setting, in addition to our observation about the larger search space (line 194). We adapt PyG’s GNNexplainer implementation for hypergraphs and otherwise use defaults, thresholding edge weights at 0.5 as suggested, and increasing epochs to 400 to match the setting for SHypX. The current PyG implementation is too slow to produce many explanations for large graphs, so currently we just report results for CoauthorCora, Zoo, and all synthetic datasets:

| | | fid-acc | fid-kl | fid-tv | fid-xent | size | density  |
|---|---|---|---|---|---|---|---|
|
| H-RandHouse | GNNexplainer | 0.54 | 0.85	| 0.44 | 1.39 | 3.3 | 0.10 |
| | SHypX | 0.01 | 0.04	| 0.06 | 0.59 | 9.2 | 0.19 |
|
| H-CommHouse | GNNexplainer | 0.66 | 3.12 | 0.67 | 3.27 | 4.1 | 0.08 |
| | SHypX | 0.00 | 0.02 | 0.03 | 0.18 | 9.2 | 0.20 |
|
| H-TreeCycle | GNNexplainer | 0.19 | 0.69 | 0.26 | 0.75 | 2.1 | 0.07 |
| | SHypX | 0.00 | 0.01 | 0.01 | 0.07 | 5.6	| 0.22 |
|
| H-TreeGrid | GNNexplainer | 0.75 | 1.59 | 0.64 | 1.78 | 3.3	| 0.09 |
| | SHypX | 0.01 | 0.02	| 0.04 | 0.22 | 15.1 | 0.45 |
|
| CoauthorCora | GNNexplainer | 0.31 | 0.60	| 0.26 | 0.66 | 1.6 | 0.20 |
| | SHypX | 0.00 | 0.00 | 0.00 | 0.07 | 2.1 | 0.29 |
|
| Zoo | GNNexplainer | 0.02 | 0.07	| 0.04 | 0.25 | 21.8 | 0.02 |
| | SHypX | 0.03 | 0.01 | 0.01 | 0.19 | 6.7	| 0.01 |

We see that GNNexplainer significantly underperforms SHypX (whose results we reproduce here), and in some cases also the other baselines (refer to Table 1 and Table 2 in manuscript). Note that we actually gave GNNexplainer a comparable or more generous size budget than SHypX (size penalty of 0.005 vs. 0.05 on synthetic datasets and 0.005 on real datasets). GNNexplainer explanations are small because few node-hyperedge links are given high attributions.

To better understand where GNNexplainer is struggling, we also look into the fidelity metrics on the pre-thresholding ‘soft mask’. Here, the whole computational subhypergraph is present but with edge weights. We see that this ‘soft mask’ is more faithful, but still generally worse than SHypX. The thresholding to obtain a subhypergraph introduces further error. From these observations, we hypothesize that for hyperGNNs 1) GNNexplainer tends to get stuck in local minima, unlike our sampler which introduces some randomness through Gumbel-Softmax, and 2) GNNexplainer’s entropy penalty fails to make the ‘soft mask’ seen during optimization sufficiently similar to the thresholded final explanation. SHypX avoids this problem by discrete sampling.

| | | fid-acc | fid-kl | fid-tv | fid-xent |
|---|---|---|---|---|---|
| H-RandHouse | GNNexplainer 'soft' | 0.48 | 0.68 | 0.40 | 1.22 |
| CoauthorCora | GNNexplainer 'soft' | 0.00 | 0.01	| 0.01 | 0.07|
| Zoo | GNNexplainer 'soft' | 0.00 | 0.04 | 0.05 | 0.22 |

[1] Pope, Phillip E., et al. "Explainability methods for graph convolutional neural networks." Proceedings of the IEEE/CVF conference on computer vision and pattern recognition. 2019.

[2] Huang, Qiang, et al. "Graphlime: Local interpretable model explanations for graph neural networks." IEEE Transactions on Knowledge and Data Engineering 35.7 (2022): 6968-6972.

---

### Meta-Review · Area_Chair_fv1T · 2024-12-21

**Metareview:**

The paper introduces a model-agnostic post-hoc explainer for hyperGNNs, providing both local and global explanations. For local explanations, it identifies important sub-hypergraphs to explain individual predictions using Gumbel-Softmax sampling, balancing faithfulness and conciseness. On the other hand, for global explanations, it derives concepts by clustering network representations and visualizing representative examples.

However, despite the rebuttal discussion, the reviewers remain unconvinced about the comprehensiveness of the empirical benchmarking and the validity of claims regarding human interpretability. As a result, the paper is considered not ready for publication in its current form.

**Additional Comments On Reviewer Discussion:**

The paper underwent discussion during the rebuttal phase. The reviewers remained unconvinced the comprehensiveness of the empirical benchmarking to more recent explainability methods, and verification of the claims related to human-interpretability.

---

### Decision · Program_Chairs · 2025-01-22

Reject